# Small extracellular vesicles containing arginase-1 suppress T-cell responses and promote tumor growth in ovarian carcinoma

Malgorzata Czystowska-Kuzmicz[1], Anna Sosnowska[1,2], Dominika Nowis[1,3,4], Kavita Ramji[1], Marta Szajnik[5,6], Justyna Chlebowska-Tuz[3], Ewa Wolinska[7], Pawel Gaj[8], Magdalena Grazul[1], Zofia Pilch[1], Abdessamad Zerrouqi[1], Agnieszka Graczyk-Jarzynka[1], Karolina Soroczynska[1], Szczepan Cierniak[9], Robert Koktysz[9], Esther Elishaev[10], Slawomir Gruca[1], Artur Stefanowicz[11], Roman Blaszczyk[12], Bartlomiej Borek[12], Anna Gzik[12], Theresa Whiteside[10,13] & Jakub Golab[1,14]

Tumor-driven immune suppression is a major barrier to successful immunotherapy in ovarian carcinomas (OvCa). Among various mechanisms responsible for immune suppression, arginase-1 (ARG1)-carrying small extracellular vesicles (EVs) emerge as important contributors to tumor growth and tumor escape from the host immune system. Here, we report that small EVs found in the ascites and plasma of OvCa patients contain ARG1. EVs suppress proliferation of CD4[+] and CD8[+] T-cells in vitro and in vivo in OvCa mouse models. In mice, ARG1-containing EVs are transported to draining lymph nodes, taken up by dendritic cells and inhibit antigen-specific T-cell proliferation. Increased expression of ARG1 in mouse OvCa cells is associated with accelerated tumor progression that can be blocked by an arginase inhibitor. Altogether, our studies show that tumor cells use EVs as vehicles to carry over long distances and deliver to immune cells a metabolic checkpoint molecule – ARG1, mitigating anti-tumor immune responses.

[1] Department of Immunology, Medical University of Warsaw, Warsaw 02-097, Poland. [2] School of Molecular Medicine, Medical University of Warsaw, Warsaw 02-091, Poland. [3] Laboratory of Experimental Medicine, Center of New Technologies, University of Warsaw, Warsaw 02-097, Poland. [4] Genomic Medicine, Medical University of Warsaw, Warsaw 02-097, Poland. [5] Holy Family Obstetrics and Gynecology Hospital, Warsaw 02-544, Poland. [6] Institute of Mother and Child, Obstetrics and Gynaecology Clinic, Warsaw 01-211, Poland. [7] Department of Pathology, Medical University of Warsaw, Warsaw 02-142, Poland. [8] Laboratory of Human Cancer Genetics, Center of New Technologies, University of Warsaw, Warsaw 02-097, Poland. [9] Department of Gynecology and Gynecologic Oncology, Military Institute of Medicine, Warsaw 04-141, Poland. [10] Department of Pathology, University of Pittsburgh, School of Medicine, Pittsburgh, 15123 PA, USA. [11] Department of Gynecology and Obstetrics, "Praski" Hospital, Warsaw 03-401, Poland. [12] OncoArendi Therapeutics, Warsaw 02-089, Poland. [13] Department of Immunology and Otorhinolaryngology, UPMC Hillman Cancer Center, Pittsburgh, 15213 PA, USA. [14] Centre of Preclinical Research, Medical University of Warsaw, Warsaw 02-091, Poland. Correspondence and requests for materials should be addressed to J.G. (email: jakub.golab@wum.edu.pl)

Epithelial ovarian cancer (OvCa) is the most lethal gyneco-logic cancer in developed countries[1]. Initially most OvCa patients respond to standard platinum-based chemotherapy that follows debulking surgery. However, due to resistance of tumor cells to chemotherapy, the majority of patients relapse and die within several years after initial remission[2,3]. Novel therapeutic approaches to OvCa are an unmet clinical need. Immunotherapy is considered to be a promising addition to chemotherapy, as OvCa cells express immunogenic tumor-associated antigens, which could be potential targets for specific immune responses[4]. However, although immunotherapeutic approaches, especially immune check-point inhibitors proved effective in the treatment of several tumor types including melanoma or non-small cell lung cancer (NSCLC), their antitumor efficacy appears to be modest in OvCa patients[5,6]. This limited efficacy might be partly explained by the highly immunosuppressive tumor microenvironment (TME) of ovarian carcinomas[7]. A number of immunoregulatory mechanisms have been identified that seem to be responsible for tumor resistance to immune therapies and for the often unsuccessful clinical responses to anticancer vaccines in OvCa[8]. These include downregulation of tumor-associated antigens and antigen-presenting machinery[9], induction of suppressive cells including T regulatory cells[10], B7H4+-macrophages[11] and tolerance-inducing plasmacytoid dendritic cells, production of immunosuppressive cytokines, such as IL-10 and TGFβ[12] and induction of oxidative stress[13], among others. Recent studies have also demonstrated that specific enzymes present in the TME are able to inhibit the immune response by limiting amino acid availability[14]. Among them are arginases that catalyze degradation of semi-essential L-arginine to L-ornithine and urea. Besides their fundamental role in the hepatic urea cycle, arginases have been shown to downregulate expression of T-cell receptor (TCR)-associated CD3ζ and ε chains, the critical components of the TCR-signaling complex, thereby impairing T-cell functions[15,16]. Moreover, depletion of L-arginine from the microenvironment arrests T-cell cycle progression and inhibits IFN-γ production[17,18]. Arginase activity also leads to the down-modulation of the expression of MHC class II molecules that are necessary for antigen presentation[19].

There are two arginase isoforms (ARG1 and ARG2), catalyzing the same biochemical reaction, but differing in subcellular localization, expression, and regulation. ARG1 is a cytosolic protein, while ARG2 is mainly localized in the mitochondria[20]. High arginase levels, either ARG1 or ARG2, have been reported in several cancer types, including breast cancer[21], NSCLC[22], head and neck squamous cell carcinoma[23], renal carcinoma[24], colorectal cancer[25], skin cancer[26], and cervical cancer[27]. Arginases are mainly produced by myeloid-derived suppressor cells (MDSCs) that are highly enriched in the TME, and the role of ARG1-expressing MDSCs in altering T-cell responses in patients with cancer has been well established[28]. Nonetheless, an increasing number of recent studies detected arginases in tumor cell lines or primary tumors, e.g. in prostate cancer[29], neuroblastoma[30], and acute myeloid leukemia[31]. However, the expression of arginase and its immunomodulatory effects in OvCa have not been described so far.

A recent study reported that nanometer-sized membrane-encapsulated extracellular vesicles (EVs) isolated from the ascites of OvCa patients and identified as exosomes, suppressed TCR-dependent nuclear translocation of NFκB and NFAT, CD69 and CD107a upregulation, and inhibited T-cell proliferation and cytokine production[32]. This report and the finding that OvCa-derived membrane vesicles can suppress CD3ζ levels in T-cells[33] provided the rationale for a more detailed investigation of the arginase expression in OvCa cells, as well as its presence and

function in OvCa-derived EVs. The overriding objective was to determine whether OvCa-derived EVs may contain ARG1 and suppress anti-tumor functions of T-cells thus providing the tumor with an advantage to escape from the host immune system.

Here we report that OvCa cells release ARG1 in small EVs and we investigate the influence of ARG1+ EVs on the antitumor effector mechanisms of immune response. We show that EVs distribute ARG1 from tumor cells to antigen-presenting cells in secondary lymphoid organs, suppressing antigen-specific T-cell proliferation and activation. We correlate high ARG1 expression in primary tumors and increased ARG1 activity in plasma with worse prognosis in OvCa patients. In an OvCa mouse model, we show that blocking arginase activity mitigates ARG1-driven tumor progression. Collectively, this study provides the first evidence for the role of ARG1+ EVs in the formation of an immunosuppressive microenvironment in OvCa.

## Results

**ARG1 expression in primary OvCa.** We first evaluated protein levels of ARG1 in established OvCa cell lines. These were relatively high in all tested cell lines as evaluated by immunoblotting (Fig. 1a) and, after cell permeabilization, by flow-cytometry (Fig. 1b). ARG1 was also expressed in tumor cells isolated from the ascites of an OvCa patient (Fig. 1a, b). Next, immunohistochemistry was used to evaluate ARG1 expression in 84 primary ovarian tumors and in normal ovary epithelial tissues that were used as controls. Supplementary Table 1 lists clinicopathologic characteristics of the patient cohort. All tumor specimens showed predominantly cytoplasmic staining for ARG1 of variable intensity (Fig. 1c). Strong staining for ARG1, with ARG scores up to 280 (see "Methods" section for the ARG score definition) was observed in 9 patients (10.71% of the tumor samples; Supplementary Table 2). In 47 patients (55.95% of the samples) a moderate ARG1 expression was observed with ARG scores of 100–180 and 28 (33.33%) of the tumors showed weak ARG1 expression (ARG scores below 100) with up to 50% of tumor cells stained weakly, and rest of the cells scored as negative. No ARG1 expression was detected in epithelial or stromal cells of the normal ovary.

Analysis of publicly available gene expression data sets of primary OvCa tumors from The Cancer Genome Atlas (TCGA) indicated that high ARG1 expression in the tumor corresponded to worse prognosis. In the analysis of a cohort of 215 patients over 50 years old, those with the lowest ARG1 expression (lowest quartile) had a significantly better overall survival (OS) than patients with the highest ARG1 expression (Cox proportional hazards model $P = 0.0246$, Fig. 1d). Analysis of another transcriptomic data set of 75 patients with serous ovarian carcinoma (Pamula-Pilat-101 MAS 5.0-u133p2; GEO accession #GSE63885) indicated that patients with low ARG1 gene expression had a significantly longer OS (Supplementary Fig. 1a, $P = 0.049$, log-rank test) and progression-free survival (Supplementary Fig. 1b, $P = 0.022$, log-rank test) than patients with high ARG1 expression.

Next, we have measured arginase activity in the plasma samples obtained from 81 untreated OvCa patients. Clinicopathologic characteristics of the patient cohort are listed in Supplementary Table 3. Arginase activity was significantly higher in the plasma of patients with stage II and III tumors (mean activity of 9.27 and 10.74 U L$^{-1}$, respectively) relative to normal controls (mean activity of 2.29 U L$^{-1}$, $P = 0.0024$ and $P < 0.0001$, respectively; Fig. 1e and Supplementary Table 4, Kruskal–Wallis test with Dunn's multiple comparison test). Furthermore, the arginase activity correlated positively with the tumor grade, increasing from a mean activity of 5.08 U L$^{-1}$ in grade I tumors

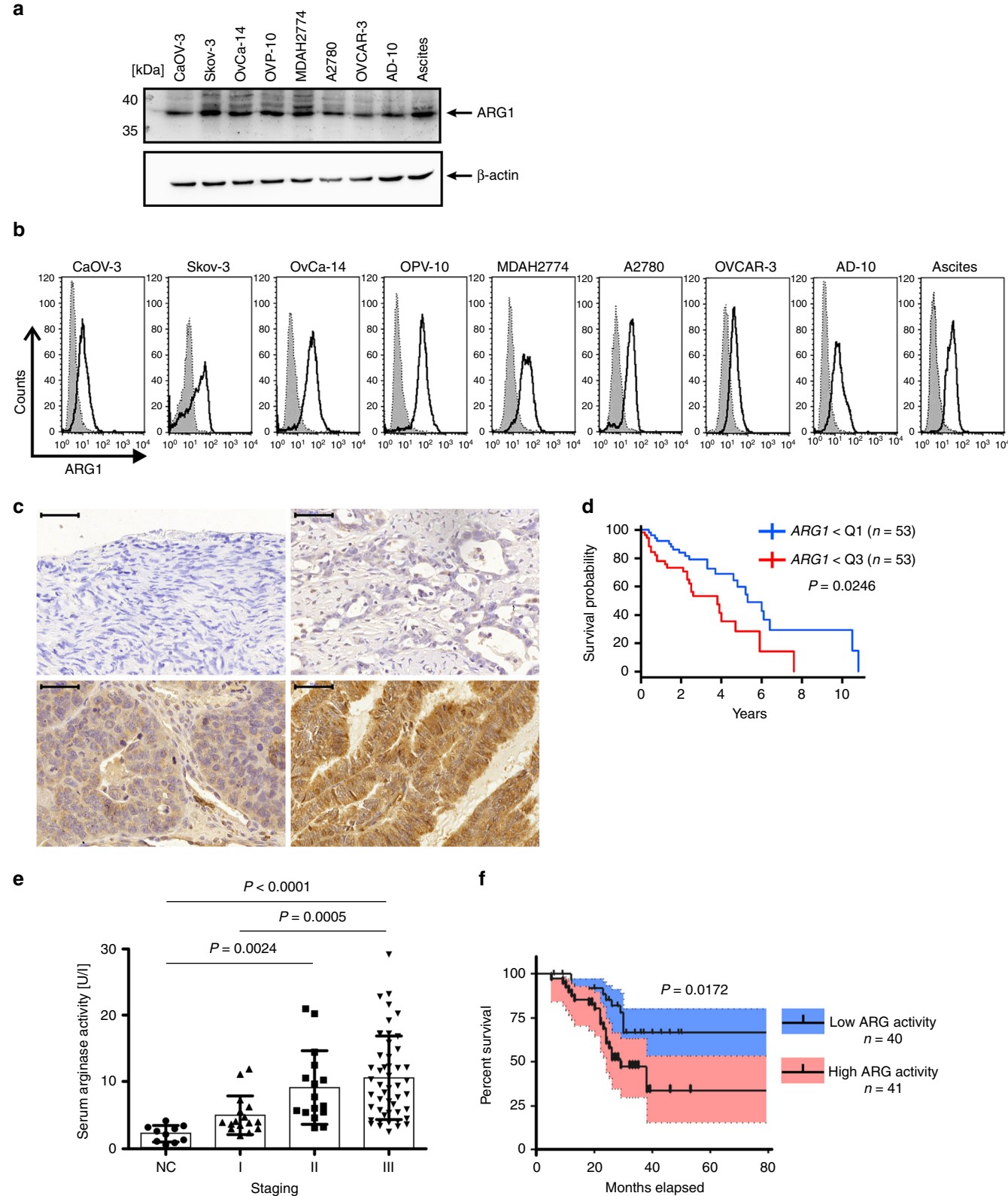

to 10.74 U L$^{-1}$ in grade III tumors. We used the log-rank test to find the point (cut-off) with the most significant (lowest $P$-value) split in high vs. low ARG1 level group according to OS. Patients with arginase activity ≥ 7.5 U L$^{-1}$ (high ARG activity) had significantly shorter OS (Fig. 1f, $P = 0.0172$, log-rank test). Altogether, we have found that increased ARG1 expression in the

primary tumor, as well as increased ARG activity in plasma correlates with poor prognosis.

**ARG1 is carried by OvCa-derived small EVs**. We and others have shown that OvCa tumors release EVs, which can be found in large quantities in the patients' plasma and ascites[32–34]. EVs

**Fig. 1** ARG1 is expressed in ovarian tumors and its levels correlate with poor prognosis. **a**, **b** ARG1 expression in OvCa cell lines and tumor cells obtained from ascites determined by Western blotting and flow cytometry, respectively. The shaded area (in **b**) represents isotype control staining, whereas the transparent one reflects ARG1 expression. **c** Representative immunohistochemistry staining of ARG1 in normal ovary (upper left) and ovarian tumor (upper right and lower panels) sections at diagnosis (bar = 50 μm). Images represent no (i), weak (ii), moderate (iii), and strong (iv) staining intensity for ARG1. **d** Kaplan–Meier curve showing higher survival probability for $n = 53$ patients demonstrating low *ARG1* transcript levels (lower quartile Q1) as compared to $n = 53$ patients with elevated (upper quartile Q3) expression levels of the gene. $P = 0.0246$ has been computed with the Cox proportional hazards model with age, clinical stage, and tumor grade included in the analysis. **e** Arginase activity in plasma of $n = 81$ OvCa (staging I–III) patients at the time of diagnosis and of 10 healthy controls (NC) determined in a colorimetric assay. Data show means ± standard deviation (SD), $*P = 0.0024$; $**P = 0.0005$; $***P < 0.0001$, Kruskal–Wallis test with Dunn's multiple comparison test. For every patient the mean activity of three independent measurements is shown. **f** Percent survival of $n = 81$ OvCa patients with high plasma arginase activity (two upper quartiles, pink) and low plasma arginase activity (two lower quartiles, blue). $P = 0.0172$ high ($\geq 7.5$ U ml$^{-1}$) vs. low ($<7.5$ U ml$^{-1}$) ARG activity, log-rank test, blue/pink shading −95% confidence interval. Source data for panels **a** and **e** are provided as a Source Data file

specifically derived from tumor cells (referred to as tumor-derived EVs, tEVs) are expected to carry a molecular signature that partly reflects that of the parental tumor cells. tEVs are enriched in immunoinhibitory molecules that may inhibit and reduce anti-tumor immune responses[35]. We therefore asked whether ARG1 can be detected in tEVs isolated from cultured OvCa cells. Immunoblotting revealed that ARG1 is present in the lysates of OvCa cell lines, as well as in EVs isolated by sequential centrifugation from supernatants of these cells (Fig. 2a). We performed Nano Sight analyses, transmission electron microscopy (TEM), and immunoblots for endocytic proteins to closely characterize the obtained EVs. EVs produced by the representative OvCa cell line Skov-3 had the mean particle diameter of 128 nm and their concentration was $9.45 \times 10^8$ particles/ml of cell supernatant (Supplementary Fig. 2a). Next, EVs isolated from the ascites of OvCa patients were examined for ARG1 content. We have detected ARG1$^+$ EVs in the ascites of OvCa patients but not in the fluid obtained from benign ovarian cysts (normal controls, NC) (Fig. 2b). Furthermore, we confirmed the presence of Tsg101 or CD63 (at least in some cases) in the isolated vesicles. Considering poor quality of immunoblots for CD63 with the available antibody we have used beads coated with antibodies targeting either CD9, CD63, or CD81 tetraspanins to immune-capture EVs pre-enriched by size-exclusion chromatography (SEC) from ascites of six different OvCa patients and to analyze the presence of the three mentioned tetraspanins in the obtained EV-subtypes (Supplementary Fig. 2c). TEM confirmed the typical morphological characteristics of small EVs: round to oval shaped vesicles surrounded by a double membrane and ranging in size from 60 to 120 nm. Immunogold staining for ARG1 confirmed the presence of this enzyme in EVs (Fig. 2c). Examination by Nano Sight (Supplementary Fig. 2b) of a representative patient sample gave the mean particle diameter of 125 nm and the concentration of $4.18 \times 10^9$ particles per ml of ascites. Since tumor-derived EVs in ascites represent only a small fraction of the total number of EVs present, we have developed an immunoaffinity-based capture method using microbeads coated with anti-EpCAM antibody. EVs directly isolated from ascites with EPCAM-beads were ARG1$^+$ and showed markers associated with EVs (Supplementary Fig. 2d). The mean size of these vesicles measured in qNano was comparable to the size of EVs isolated by SEC (Supplementary Fig. 2e). Using sequential centrifugation, we retrieved EVs from 49 patients with grade III OvCa and detected ARG1 in 29 of cases. We also isolated small EVs from benign cysts fluid of 9 patients and used these EVs as NC. OvCa patients had ~2.5 times more EVs (measured as the total protein concentration in the EV lysate, Fig. 2d, left) than normal controls and were characterized by a higher expression of Tsg101, a characteristic marker of small EVs[36] (Fig. 2d, middle), as well as slightly higher, but statistically insignificant ($P = 0.1327$, unpaired *t*-test with Welch's correction) ARG1 levels (Fig. 2d, right).

Next, by measuring the ability of EVs to convert L-arginine to urea, we confirmed that ARG1 in EVs derived either from OvCa cell line supernatants or the OvCa patients' ascites was enzymatically active. Arginase activity in the lysates of all tested OvCa cell lines ranged from 4.82 to 26.9 U/g of total protein (Fig. 2e), while the EVs isolated from the ascites of OvCa patients had enzymatic ARG activity in the range of 0.305–6.514 mU/ml of ascites (Fig. 2f). The enzymatic ARG activity of ascites EVs was higher than the activity in EVs isolated from fluid of benign cysts (Fig. 2f, g).

**EVs with ARG1 suppress peripheral T-cells in OvCa patients.** Since patients with OvCa have increased arginase activity in the TME and in the peripheral circulation, we questioned whether ARG1$^+$ small EVs are detectable in the plasma of OvCa patients. ARG1 was detected in the lysates of plasma EVs by immunoblotting. These EVs contained more ARG1 than EVs obtained from the plasma of patients with benign disease. The relative levels of ARG1 in the plasma EVs of OvCa patients corresponded to increased plasma arginase activity (Fig. 3a).

To determine whether the increased ARG1 in plasma EVs of OvCa patients may associate with the ability of peripheral T-cells to proliferate, we concomitantly measured arginase activity and assayed T-cell proliferation and CD3ζ levels in 14 patients and 5 controls with benign ovarian cysts. Ex vivo T-cell proliferation was induced with anti-CD3/CD28-coupled beads. Representative histograms of T-cell proliferation are shown in Supplementary Fig. 4a. Only 4/14 OvCa patients had normal T-cell proliferation. In seven patients we observed almost no proliferating CD8$^+$ T-cells, and in the remaining three patients, the proliferation was moderately inhibited (Fig. 3b). The observed proliferation impairment of CD8$^+$ cells correlated positively with reduced CD3ζ-chain expression in these cells (Fig. 3c and Supplementary Fig. 3b). Reduced CD8$^+$ T-cell proliferation and decreased CD3ζ expression levels correlated with increased ARG activity in the plasma (Fig. 3d, e). Similar results were also observed with CD4$^+$ T-cells (Supplementary Figs. 3 and 4). In contrast, T-cells of the five patients with benign conditions (endometrial cyst or uterine myoma) showed normal proliferation and stable levels of CD3ζ expression. Moreover, the measured arginase activity was lower in ascitic fractions after isolation of EVs as compared to full ascites (Supplementary Fig. 5a) and ascitic fractions remaining after EV isolations were less effective in inhibiting T-cell proliferation as compared with full ascites (Supplementary Fig. 5b). Altogether, these results indicate that increased ARG content in the plasma EVs of OvCa patients correlates with decreased CD3ζ levels and impaired proliferation of peripheral T-cells.

**ARG1 in EVs suppresses T-cell proliferation in vitro.** Then, we sought to investigate the effects of ARG1-containing EVs obtained from OvCa-patients ascites on the proliferation of

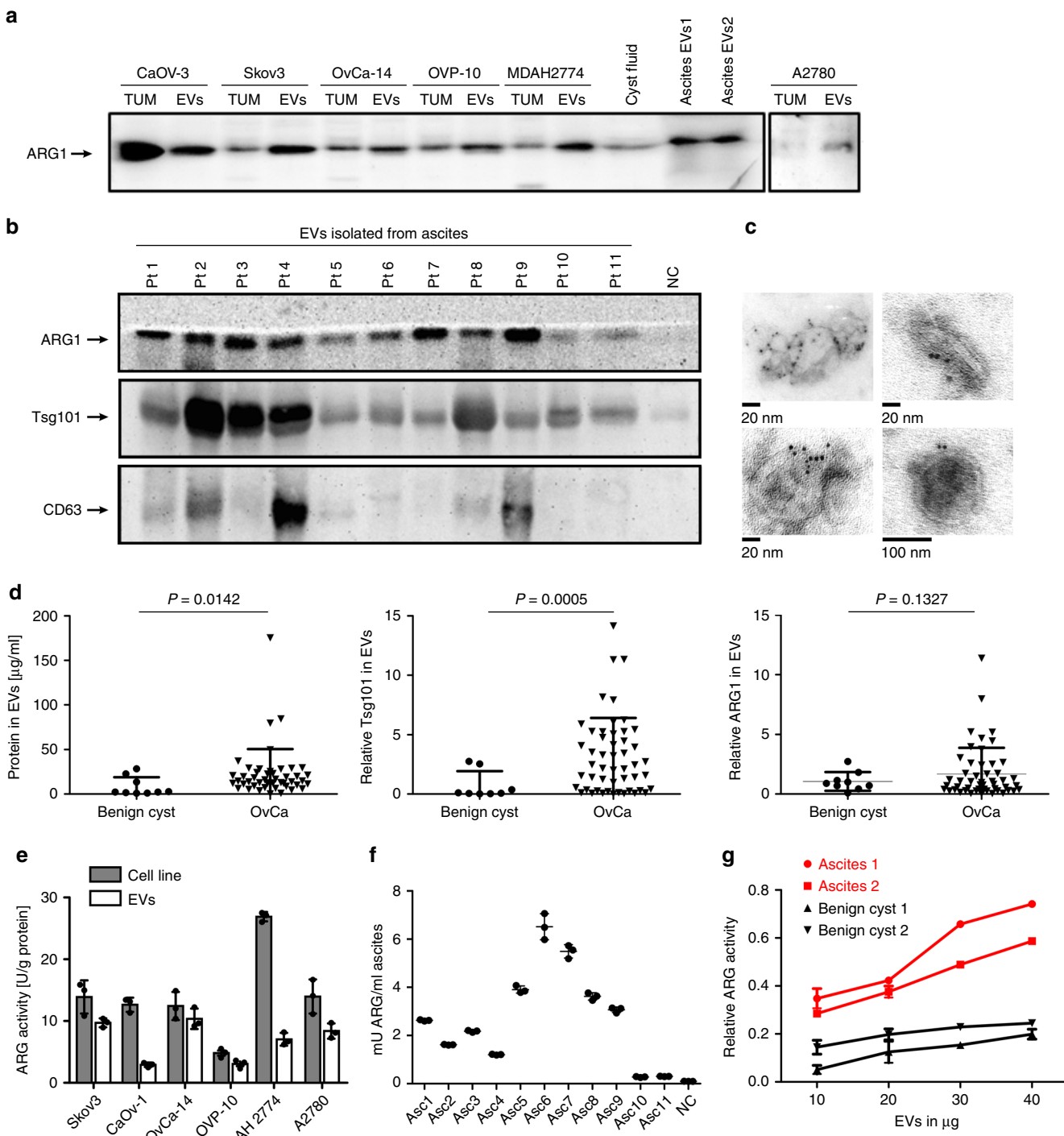

**Fig. 2** OvCa-derived EVs contain enzymatically active ARG1. **a** ARG1 content in EVs and the parental OvCa cell lines lysates (TUM) as well as in EVs isolated from the cyst fluid and ascites fluid of two OvCa patients (EV1, EV2) determined by Western blotting. Equal amounts of protein (30 µg) were loaded per lane. **b** Representative Western blot for ARG1 and typical exosomal markers in EVs isolated from ascites of $n = 11$ OvCa patients. EVs isolated from ovarian fluid served as normal control (NC). Equal amounts of protein (30 µg) were loaded per lane. **c** Representative TEM images of whole-mounted OvCa patient-derived tEVs. Dots indicate immunogold (6 nm gold particles) labeling of ARG1. **d** Protein concentration measured with BCA assay, Tsg101 and ARG1 levels in EVs isolated from 2 ml of ascites of OvCa patients ($n = 47–50$) and from 2 ml of benign cysts fluid ($n = 7–9$). Relative Tsg101 and ARG1 content was determined by densitometric analysis of Western blots. Data refers to means ± SD, $P$ values were calculated with unpaired $t$-test with Welch's correction. **e** Arginase activity in OvCa cell line lysates and in the corresponding EVs determined by measuring ʟ-arginine conversion to urea in a colorimetric assay. Data show means ± SD, $n = 3$. **f** Arginase activity in EVs isolated from $n = 11$ OvCa patients' ascites (Asc) and benign cyst fluid (NC) calculated per ml of starting fluid volume. Data show means ± SD, $n = 3$. **g** Arginase activity as a function of amount of EVs in $n = 2$ OvCa patients ascites-isolated tEVs and EVs isolated from $n = 2$ benign cyst fluid. Data show means ± SD. Source data for data in panels **a**, **b**, **d–g** are provided as a Source Data file

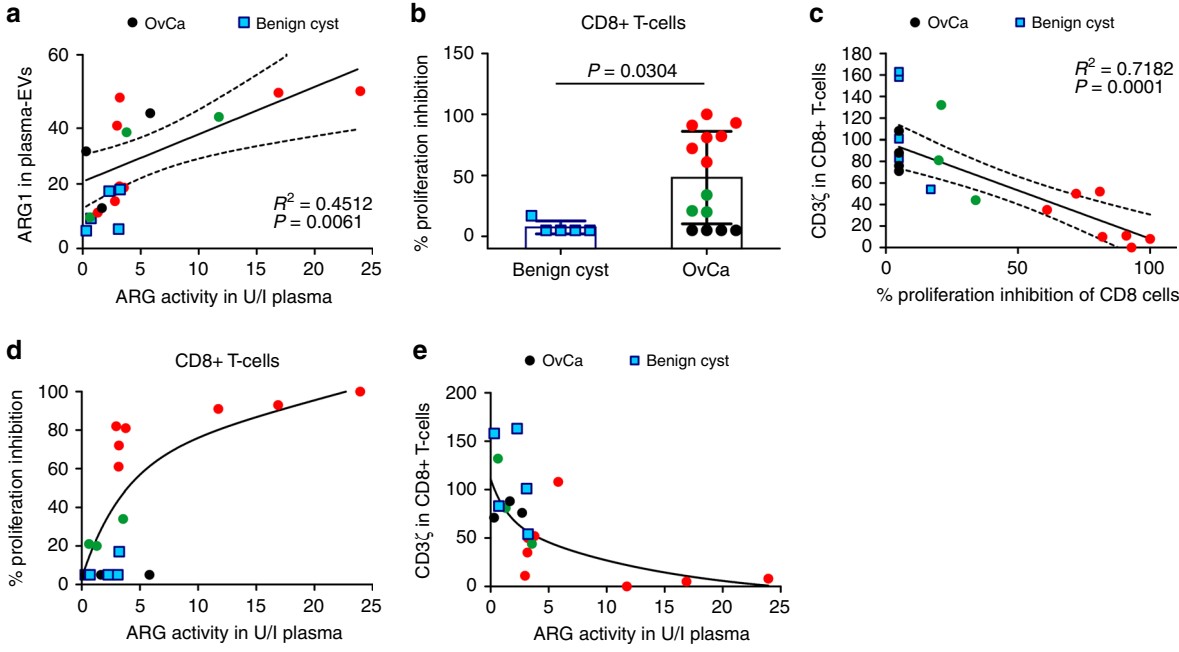

**Fig. 3** ARG1-containing EVs contribute to systemic immune suppression in OvCa patients. **a** Correlation of relative expression of ARG1 in plasma-derived EVs (isolated from $n = 14$ OvCa patients [circles] and $n = 5$ patients with benign cyst of the ovary [squares]) and plasma ARG activity. $R^2$ value and $P$ values calculated with GraphPad Prism 6.0. Dotted lines mark 95% confidence intervals. **b–e** Black circles mark OvCa samples with no CD8+ T-cell proliferation inhibition ($n = 4$), green circles ($n = 3$)—moderate proliferation inhibition, and red circles ($n = 7$)—complete or nearly complete proliferation inhibition. Blue squares mark benign cyst samples. **b** Proliferation of peripheral blood CD8+ T-cells in $n = 14$ OvCa patients and $n = 5$ patients with benign ovarian cyst normalized to the mean proliferation of $n = 5$ healthy controls. Data show means ± SD, $P$-value calculated with Mann–Whitney $U$-test. **c** Correlation of the peripheral blood CD8+ T cell percentage of proliferation inhibition and CD3ζ levels evaluated in flow cytometry. $R^2$ value and $P$ value calculated with GraphPad Prism 6.0. Dotted lines mark 95% confidence intervals. **d** Percentages of peripheral blood CD8+ cells proliferation inhibition as a function of ARG activity in plasma of $n = 14$ OvCa patients and $n = 5$ patients with benign cyst of the ovary. **e** CD3ζ levels evaluated in flow cytometry as a function of ARG activity in the plasma of $n = 14$ OvCa patients and $n = 5$ patients with benign cyst of the ovary. Source data are provided as a Source Data file

normal T-cells in vitro. Human peripheral blood CD4+ and CD8+ T-cells were stimulated with anti-CD3/CD28-coupled beads and incubated with EVs isolated from 44 OvCa patients' ascites. As controls we have used EVs isolated from the fluid of seven benign ovarian cysts (cystic fluid EV, CFEV). CFEVs did not or only marginally suppressed proliferation of the T-cell subsets. In contrast, a significant suppression of T-cell proliferation was observed using EVs isolated from OvCa patients (Fig. 4a, b). Notably, EVs isolated from 18 out of 43 (42%) and 18 out of 44 (43%) of OvCa patients very strongly (by over 70%) inhibited the proliferation of CD4+ and CD8+ T-cells, respectively (Fig. 4b upper panels). EVs from 8 (19)% and 12 (27%) of the patients had no effect (inhibition by <10%) on the proliferation of CD4+ or CD8+ T-cells, respectively. Along with inhibition of T-cell proliferation we also observed a decrease in CD3ζ levels in both T-cell subsets (Fig. 4b lower panels). The observed CD3ζ downregulation corresponded to the T-cell proliferation inhibition and tended to be stronger with EVs isolated from OvCa patients than with CFEV. EVs suppressed both CD4+ and CD8+ T-cells in a dose-dependent manner (Fig. 4c). Furthermore, we observed that, at least in some cases, the specific inhibition of ARG1 with arginase inhibitor (OAT-1746)[37] or addition of excess L-arginine to the culture medium partially reversed the effects of patients'-derived EVs (Fig. 4d).

**ARG1+ EVs are internalized by DCs**. To further delineate immunoregulatory mechanisms of ARG1+ EVs, we have used a murine ID8 OvCa model. Since parental ID8 tumor cells do not express ARG1, we overexpressed V5-tagged murine ARG1

(ID8-ARG1-V5) using lentiviral transduction system. As controls we used empty vector-transduced ID8 cells (ID8-pLVX). Small EVs isolated from the supernatants of ID8-ARG1-V5 cells (EVs-ARG1) contained enzymatically active ARG1 (Supplementary Fig. 6a), the presence of which could be specifically detected by immunoblotting with an anti-V5-tag antibody (Supplementary Fig. 6b). Confocal microscopy showed that PKH67-stained EVs-ARG1 were internalized by murine bone marrow-derived dendritic cells (BMDCs) at 37 °C, but not at 4 °C (Fig. 5a) and EV-derived ARG1 was specifically detected by Western blotting in DCs lysates (Fig. 5b). The EV-associated ARG1 was functionally active, as induction of CD8+ and CD4+ T-cell proliferation with anti-CD3/CD28-coupled beads was inhibited by DCs co-incubated with EVs-ARG1 (Fig. 5c). EVs-ARG1 also reduced the expression of CD3ε in both CD8+ and CD4+ T-cells (Fig. 5d). The control EVs isolated from ID8-pLVX cells (EVs-pLVX) had no influence on T-cell proliferation. Similarly, EVs-ARG1 inhibited the OVA-peptide-specific proliferation of DC-primed OT-I T-cells (Fig. 5e). In all experimental settings, the suppressive effects of EVs-ARG1 were completely abrogated by addition of the arginase inhibitor OAT-1746, confirming the role of ARG1 in EV-mediated suppression of T-cell proliferation.

**ARG1+ EVs suppress antigen-specific T-cells in vivo.** Antigen-specific T-cell proliferation is triggered in secondary lymphoid organs. Thus, we have developed an animal model to investigate the effects of OvCa-derived tEVs on T-cell proliferation in the local lymph nodes (LNs). To this end, control EVs-pLVX as well as EVs-ARG1 were subcutaneously inoculated into C57BL/6

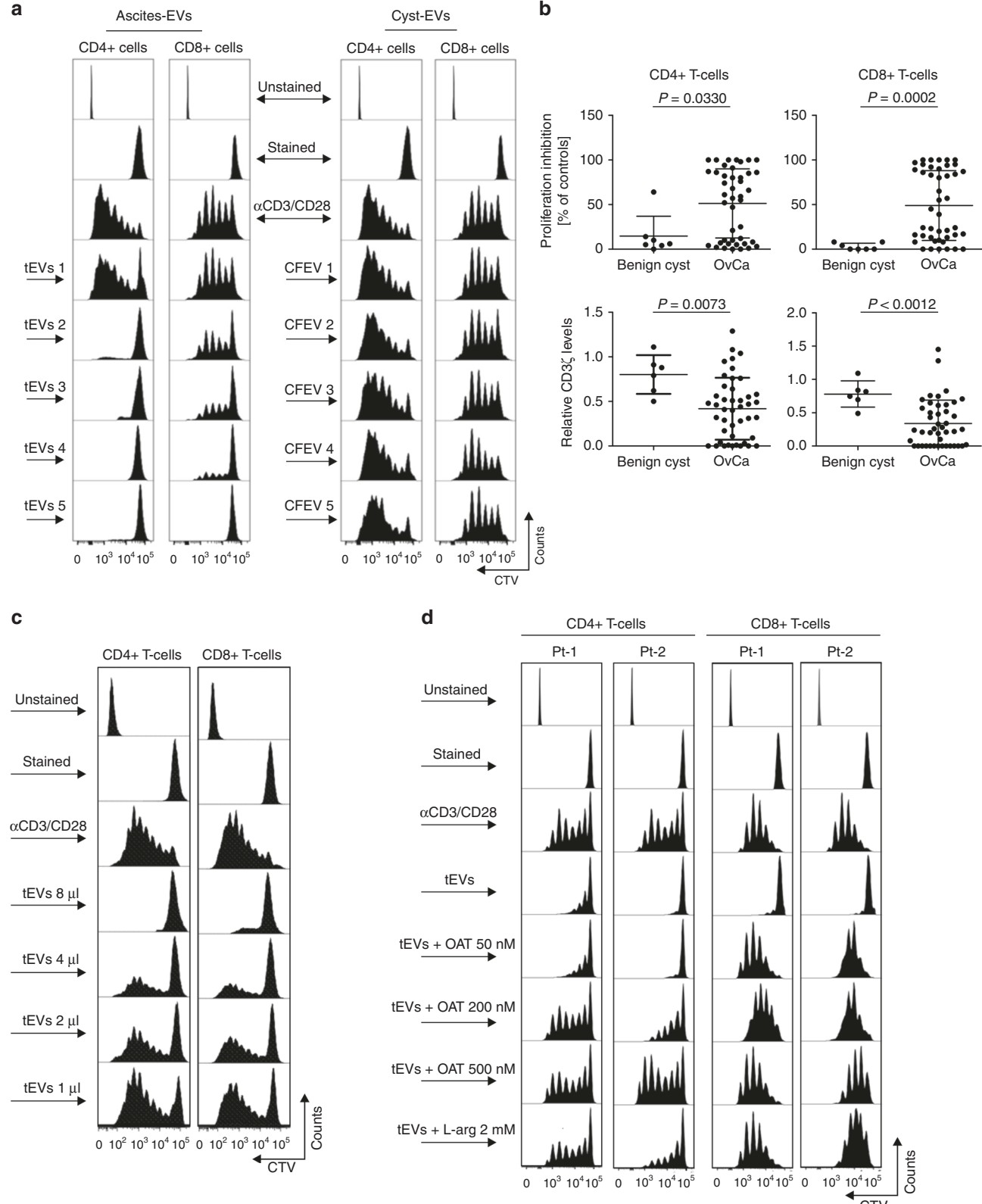

mice. Immunoblotting of lysates of local LNs demonstrated the presence of V5 tag at 4 and 24 h post tEVs inoculation (Fig. 6a), indicating that the injected tEVs were being transported to the draining LNs. Subcutaneously inoculated EVs-ARG1 significantly inhibited OVA-specific proliferation of adoptively transferred OT-I T-cells ($P = 0.0275$, Kruskal–Wallis with Dunn's multiple comparison test), whereas inoculation of control EVs-pLVX had

no effect on T cell proliferation (Fig. 6b). Notably, inoculation of recombinant mouse ARG1 (rmARG1) followed by adoptive transfer of OT-I T-cells also failed to inhibit OT-I T-cell proliferation (Supplementary Fig. 7). To further investigate whether the suppressive effects are mediated by ARG1, we performed the in vivo proliferation experiment in OAT-1746-treated mice. Pharmacological ARG1 inhibition partially reversed

**Fig. 4** EV-ARG1 is involved in the suppression of T-cell proliferation in vitro. **a** Representative proliferation histograms of αCD3/αCD28-stimulated CD8[+] and CD4[+] T cells incubated for 6 days with tumor ascites-derived EVs isolated from $n = 5$ OvCa patients (left panel, tEVs 1–5) and control EVs (CFEV1–5, right panel), isolated from $n = 5$ patients with benign cyst of the ovary. **b** Inhibition of proliferation (upper) and decrease in CD3ζ levels (lower) of peripheral blood CD4[+] (left) or CD8[+] T-cells (right) by OvCa ascitic fluid-isolated EVs ($n = 43$–44). Data show means ± SD, $P$ values for OvCa ascites vs. benign cyst fluid-isolated EVs treated group ($n = 6$–7) were calculated with Mann–Whitney $U$-test. The amount of added EVs corresponded to 2 ml of starting material **a**, **b**. **c** Representative proliferation histograms of αCD3/αCD28-stimulated CD4[+] and CD8[+] T-cells incubated with increasing amounts of tEVs, isolated from 2 ml (8 µl), 1 ml (4 µl), 0.5 ml (2 µl), or 0.25 ml (1 µl) of ascites, respectively. **d** Representative proliferation histograms of peripheral blood CD4[+] and CD8[+] T-cells incubated for 6 days with tEVs (corresponding to 2 ml of ascites) and indicated concentrations of an arginase inhibitor, OAT-1746. Cells incubated for 6 days with tEVs and 2 mM ʟ-arginine served as a positive control for ʟ-arginine-dependent reversal of T-cells proliferation inhibition. Source data for panel **b** are provided as a Source Data file

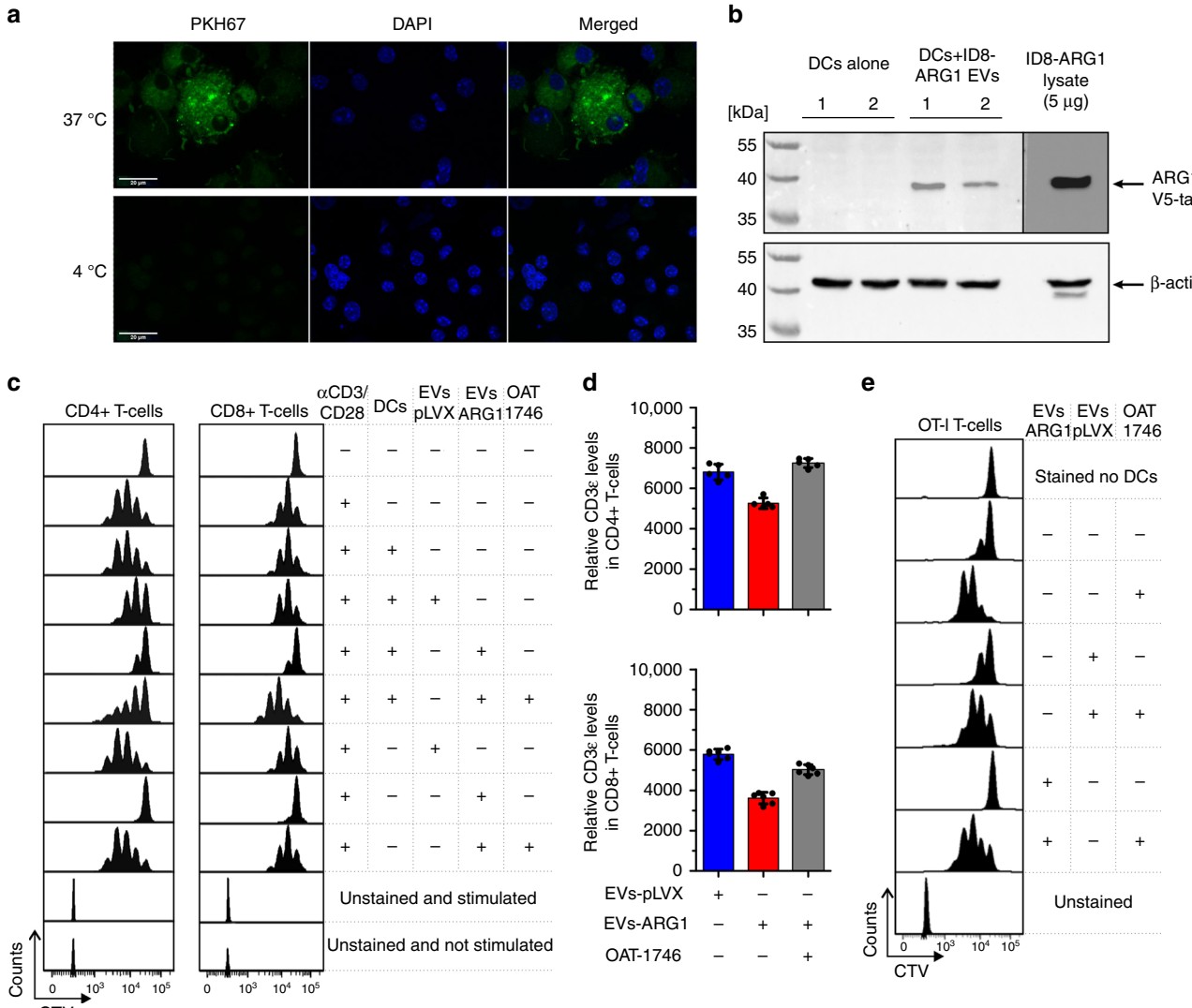

**Fig. 5** ARG1-EVs are internalized by murine bone marrow-derived DCs (BMDCs) and block DCs-primed T-cell proliferation. **a** Representative image of EVs endocytosed by BMDCs from confocal microscopy. BMDCs were incubated with 50 µg of PKH-67-stained ARG1-EVs isolated from the supernatants of ID8-ARG1-V5 murine ovarian carcinoma cell line for 4 h in 37 or 4 °C (inhibited internalization, negative control), washed and fixed. Green—PKH67-stained EVs, blue—DAPI nuclear stain. **b** Western blotting for V5-tag labeled ARG1 in DCs lysates. BMDCs were incubated with 50 µg of ARG1-EVs isolated from supernatants of ID8-ARG1-V5 murine OvCa cell line for 4 h in 37 °C, washed and lysed. ID8-ARG1-V5 (ID8-ARG1) cells lysate was used as a positive control for V5-tagged Arg1 detection. Beta-actin served as equal protein loading control. **c** Representative proliferation histograms of αCD3/αCD28-stimulated CD4[+] and CD8[+] T-cells co-cultured with BMDCs pre-incubated with 100 µg EVs isolated from the supernatants of ID8-ARG1 cells (EVs-ARG1) or ID8-pLVX (EVs-pLVX) cells for 3 days. EVs-ARG1 or EVs-pLVX with no BMDCs and/or ARG inhibitor OAT-1746 (200 nM) were added to some groups as indicated in the figure. **d** Relative CD3ε expression evaluated with flow cytometry in αCD3/αCD28-stimulated CD4[+] (upper graph) and CD8[+] (lower graph) T-cells after pre-incubation with EVs-ARG1 or EVs-pLVX (90 µg) and/or ARG inhibitor OAT-1746 (400 nM). Data show MFI of two technical repeats from $n = 3$ mice ± SD. **e** Representative proliferation histograms of SIINFEKL-specific CD8[+] T-cells (OT-I T-cells) primed with SIINFEKL peptide pulsed BMDCs. Where indicated, DCs were pre-incubated with EVs-ARG1 or EVs-pLVX (100 µg) in the presence of 200 nM ARG inhibitor OAT-1746. Source data for panels **b** and **d** are provided as a Source Data file

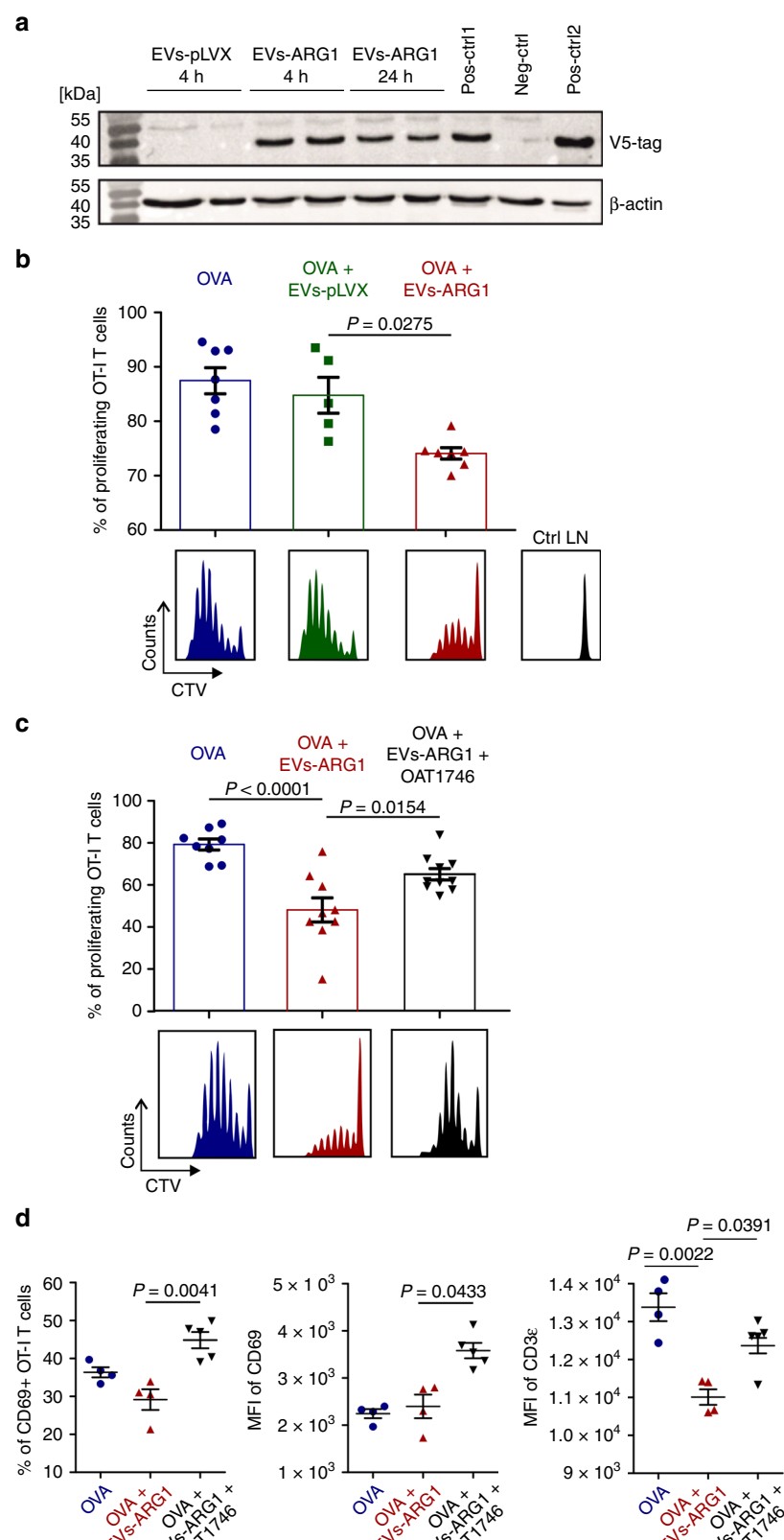

tEVs-mediated suppression of T cell proliferation (Fig. 6c). Flow cytometry analysis of OT-I cells from the draining LNs showed that ARG1$^+$ EVs significantly decreased the percentage of activated (CD69$^+$) OT-I T-cells and downregulated CD3ε expression in these cells. Upon ARG1 inhibition the CD3ε levels in OT-I cells were restored and the percentage of activated (CD69$^+$) OT-I cells and their expression of CD69 significantly increased ($P = 0.0391$, $P = 0.0041$, $P = 0.0433$, respectively, Kruskal–Wallis with Dunn's multiple comparison test), exceeding even the levels in control mice (Fig. 6d).

**Fig. 6** Subcutaneously administered EVs transfer ARG1 to local lymph nodes and block OVA-antigen-specific T-cell proliferation. **a** Right inguinal lymph nodes from C57BL/6 mice were collected 4 or 24 h post EVs (250 µg) or PBS (Neg-ctrl) inoculation, lysed, and analyzed with immunoblotting for V5 tag. Pos-ctrl1: lysate of control lymph node with ex vivo added EVs-ARG1. Pos-ctrl2: lysate from ID8-ARG1-V5 cells. β-actin served as equal protein loading control. **b–d** OVA-specific, CTV-stained OT-I T-cells were injected i.v. into C57BL/6 mice and 24 h post adoptive transfer 5 µg of OVA protein 100 µg of EVs were injected daily for the consecutive 3 days s.c. into the right thigh. Where indicated, mice received 20 mg/kg of ARG inhibitor (OAT-1746) i.p. twice a day for the 3 following days. Each experimental group consisted of $n = 4$–9 mice. After 72 h CD8$^+$ T-cells were isolated from right and left (control) inguinal lymph nodes, stained with SIINFEKL-H-2Kb tetramers and analyzed for proliferation. **b** Percentages (upper graph) and exemplary histograms (lower panel) of proliferating OT-I T cells isolated from mice immunized with ovalbumin (OVA) and injected with control EVs-pLVX or EVs-ARG1 EVs. T-cells isolated from the contralateral lymph node served as negative control (Ctrl LN). Representative experiment out of $n = 2$ is shown, data show means ± SD, P value was calculated with Kruskal–Wallis with Dunn's multiple comparison test. **c** Percentages (upper) and exemplary histograms (lower panel) of proliferating OT-I T cells isolated from mice immunized with OVA and injected with EVs-ARG1 and/or ARG inhibitor OAT-1746. Representative experiment out of $n = 2$ with similar results is shown, data show means ± SD, P values were calculated with one-way ANOVA with Bonferroni post-hoc test. **d** Percentages of CD69$^+$ T-cells (left), mean fluorescence intensity (MFI) for CD69 staining (middle) and MFI for CD3ε staining in OT-I T cells isolated from mice immunized with OVA and injected with EVs-ARG1 and/or ARG inhibitor OAT-1746. Representative experiment out of $n = 2$ is shown. Data show means ± SD, P values were calculated with Kruskal–Wallis with Dunn's multiple comparison test. Source data for panels **b–d** are provided as a Source Data file

**ARG1 promotes OvCa progression**. Since human OvCa cells express ARG1, we questioned whether ARG1 is involved in the regulation of tumor progression. Mice were inoculated i.p. with control (ID8-pLVX) or ARG1-transduced (ID8-ARG1) tumor cells and were treated with OAT-1746 or PBS starting from day 15 after inoculation of tumor cells. Tumor progression was monitored by measuring weight and waist circumference gains. Mice inoculated with ID8-ARG1 cells showed faster tumor growth as compared with control ID8-pLVX tumors (Fig. 7a). In these animals ascites formed at an earlier time point and accumulated markedly faster than in mice bearing ID8-pLVX tumors. Diffuse peritoneal dissemination of tumor cells consisting of multiple tumor nodules of 0.5–5 mm, which were dispersed on the parietal and visceral surfaces of the peritoneal cavity at 28–34 days post tumor cells inoculation was observed. Tumor nodules were particularly noticeable in the diaphragmatic peritoneum resembling human ovarian carcinoma. Control animals bearing ID8-pLVX tumors, as well as OAT-1746-treated mice displayed occasional small (0.5–2 mm) nodules on the diaphragmatic peritoneum. Serum ARG1 levels in ID8-ARG1 tumor-bearing mice increased concomitantly with the tumor growth (Supplementary Fig. 8a). The mean arginase activity in the small EV fraction obtained from the ascitic fluid collected from ID8-ARG1 tumor-bearing mice at weeks 3 ($n = 3$) and 7 ($n = 4$) was 2.08 mU/ml of ascites (Source Data File—Table 1), which is within the range of arginase activities observed in OvCa patients (Fig. 2f). Cells highly expressing ARG1-V5-tag were detected by immunoblotting in ascites 28 days post inoculation of tumor cells (Supplementary Fig. 8b). Furthermore, ARG1-V5-positive EVs were isolated from ascites of these mice (Supplementary Fig. 8c). ARG1 inhibition with OAT-1746 significantly reduced the growth of ID8-ARG1 tumors (Fig. 7a, for weight $P = 0.0130$, for waist circumference $P = 0.020$, unpaired t-test). A less pronounced, but nevertheless significant reduction of tumor growth upon OAT-1746 treatment was also observed in the control group with ARG1-negative tumors (Fig. 7a, for weight $P = 0.0065$, for waist circumference $P = 0.0093$, unpaired t-test), indicating potential blockade of arginase activity in MDSCs and TAMs, the other known sources of ARG1 in the TME. Consistent with the results obtained in vivo with the EVs-ARG1 in the adoptive transfer model in non-tumor settings, ID8-ARG1 bearing-mice had a significantly reduced percentage of activated CD69-positive CD4$^+$ (Fig. 7b left) and CD8$^+$ (Fig. 7b middle) T-cells in the peritoneal cavity relative to mice bearing control tumor cells. OAT-1746 treatment noticeably increased the percentage of activated CD4$^+$ and CD8$^+$ cells ($P = 0.0007$, $P = 0.0030$, respectively, unpaired t-test). Furthermore, in ID8-ARG1

bearing-mice up to 1% of peritoneal activated CD11c$^+$ dendritic cells stained positive for V5-tag indicating an uptake of tumor-derived ARG1 (Fig. 7b right). As we have shown in the in vitro and in vivo assays, upon internalization of ARG1 antigen-presenting DCs lose their activating potential and instead become suppressive. Due to very limited cell numbers we were not able to check the activating capacity of the V5-positive DCs in this experimental setting. Upon ARG1 inhibition the percentage of V5-positive, potentially suppressive CD11c$^+$ dendritic cells in the peritoneal cavity has decreased. Taken together, these data indicate that ARG1 expressed by tumor cells accelerates tumor progression and is a potential therapeutic target in ovarian carcinoma.

## Discussion

Arginine metabolism is one of the metabolic pathways responsible for tumor progression[38]. Since ARG1 regulates the availability of L-arginine, it is under extensive investigation as an anticancer therapeutic target[39]. However, the potential involvement of ARG1 in OvCa has received little attention to date, and in other tumors it was mostly considered in association with myeloid cells and MDSCs. Murine and human OvCa-induced MDSCs were reported to express ARG1, and murine MDSCs were shown to require ARG1 for suppression of T-cells[40]. Furthermore, higher arginase activity as compared to normal controls was detected in the plasma of OvCa patients[41] and it decreased after chemotherapy[42].

Our study is the first to report that ARG1 is expressed in OvCa and that it becomes distributed far beyond the local TME by EVs. We observed that increased arginase activity was not only confined to the TME, but was also detectable in the peripheral circulation of OvCa patients and correlated with poor survival. Therefore, we hypothesized that ARG1 can be released from tumor cells in EVs and found ARG1$^+$ small EVs in ascites, as well as in the plasma of OvCa patients. Remarkably, high arginase activity in patients' plasma along with high ARG1 content in plasma EVs correlated with lower CD3ζ expression levels and poor proliferative capacity of circulating T-cells in our patient cohort. In the following in vitro studies, we showed that patient-derived as well as OvCa cell line-derived ARG1$^+$ EVs impair the functions of human and murine T-cells by blocking their proliferation, and reducing expression levels of the CD3ζ and CD3ε chains. This transmembrane components of the TCR complex serve as essential signaling molecules in T lymphocytes, and their appropriate expression and phosphorylation are critical for T lymphocyte activities, such as proliferation and cytokine production. Numerous studies have previously demonstrated

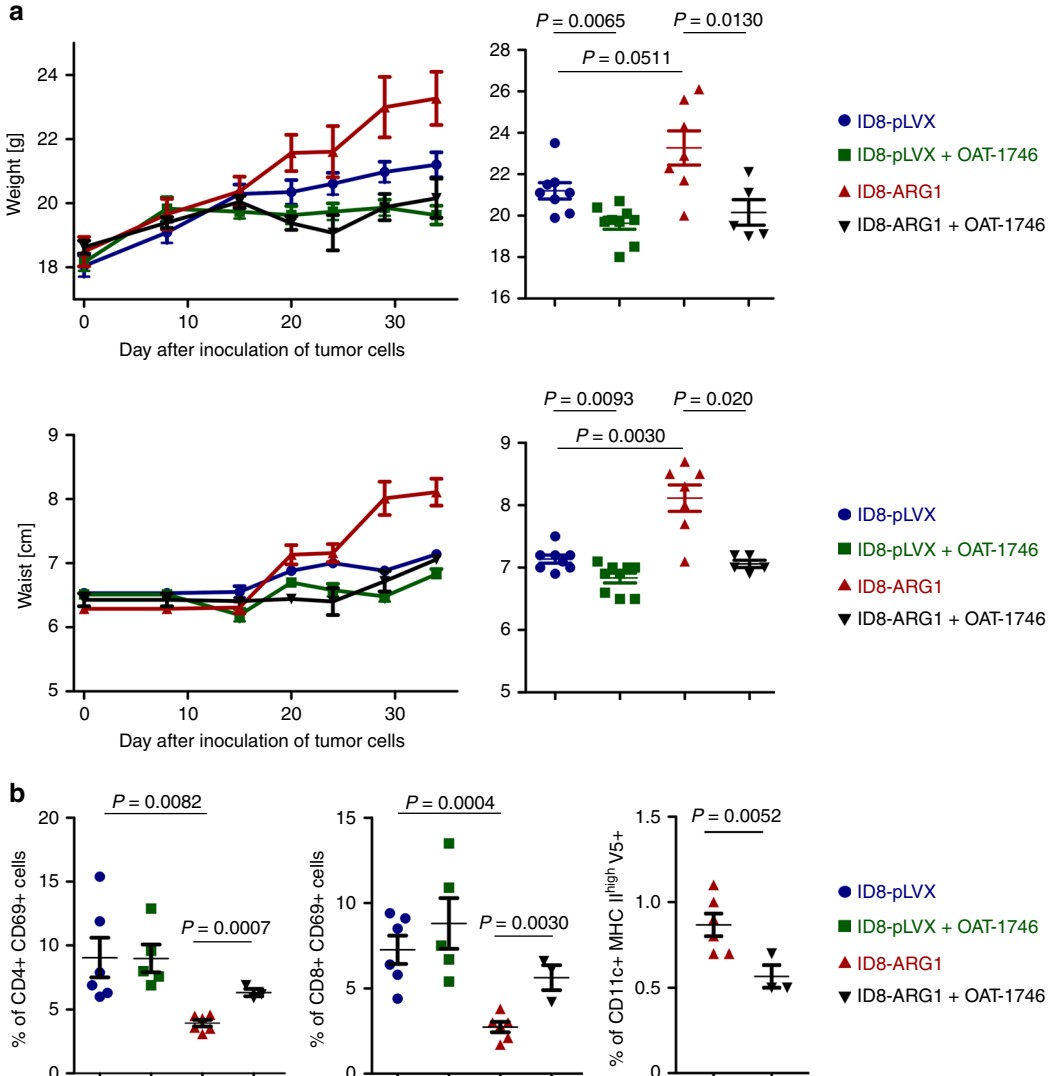

**Fig. 7** ARG1 promotes tumor growth in vivo. C57BL/6 mice were inoculated i.p. with 4 × 10⁶ ID8-*VegfA/Defb29* cells transduced with V5-tagged murine ARG1 (ID8-ARG1-V5) or the control vector (ID8-pLVX). **a** Mice were treated from day 14th after tumor inoculation with OAT-1746 or PBS i.p. twice daily and monitored for tumor development until first mice met the humane endpoint criteria described in the "Methods" section. Increase in mice weight (upper left) and waist circumference (lower left) in time compared to day 0 (day of tumor cells i.p. inoculation) as a measure of ovarian cancer progression/ ascites development. Measurements of gained weight (upper right) and percentage of gained waist circumference (lower right) on day 34 after inoculation of tumor cells. Each experimental group consisted of *n* = 5–9 mice. Data show means ± SE. **b** Mice were treated from day 14th after tumor inoculation with OAT-1746 ARG inhibitor or PBS i.p. twice daily for consecutive 2 weeks. After 2 weeks of treatment cells from ascites were isolated and analyzed by flow cytometry. Percentage of ascitic activated (identified as CD69⁺) CD4⁺ (left panel) or CD8⁺ T-cells (middle panel), as well as percentage of activated (MHCII^high) CD11c⁺ DCs positive for V5-tag (right panel) on day 28th after inoculation of tumor cells. Each experimental group consisted of *n* = 3–6 mice. Data show means ± SD; *P* values were calculated with unpaired *t*-test. Source data are provided as a Source Data file

alterations in expression and function of CD3ζ in both tumor-infiltrating lymphocytes (TILs) and peripheral blood T-cells[16,43]. While transient decrease in the CD3ζ expression level occurs normally during antigenic stimulation[44], its persistent loss, as often observed in TILs, has been correlated with reduced T-cell proliferation and cytokine production[15,45]. These alterations, along with other mechanisms, are responsible for deficient immune responsiveness of T-cells in cancer patients, including OvCa[46,47]. Importantly, the CD3ζ chain loss in T-cells of cancer patients is biologically significant, as it correlates with worse prognosis and shorter OS, as reported for OvCa[33], head and neck cancer[48], and breast cancer[49]. We and others have shown previously that tumor-derived microvesicles, including exosomes in OvCa, suppress CD3ζ chain expression of T-cells[50,51], but identified no responsible mechanism. We observed an induction of

apoptosis by tumor EVs and linked this finding with the presence of FasL or TRAIL on the exosomes surface[50,51]. Further, we have shown that EV-induced apoptosis is accompanied by caspase-3 cleavage, cytochrome c release, loss of mitochondrial membrane potential, DNA-fragmentation, and inactivation of the PI3K/Akt pathway with concomitant downregulation of anti-apoptotic proteins[50,52]. The present study defined ARG1 as an additional EV-derived component that is responsible for CD3ζ and CD3ε chain downregulation and T-cell suppression. Furthermore, we demonstrate that ARG1⁺ EVs deliver the active enzyme to the draining LNs in vivo, where vesicle ARG1 either directly inhibits T-cell proliferation by decreasing L-arginine levels or is taken up by DCs impairing their activating potential. In contrast, recombinant ARG1 even at high doses failed to inhibit specific T-cell proliferation, probably due to the fast degradation of the free

enzyme in vivo. Arginase can be produced and released from normal cells, but in the extracellular milieu, the enzyme is unstable and its circulating half-life in humans is <30 min[53]. Our results indicate, that ARG1 in EVs, in contrast to free ARG1, remains stable, possibly protected from degradation by the EVs membrane.

It has been well established that a special subpopulation of endosome-derived and tetraspanins-enriched small EVs called exosomes are involved in multiple facets of tumorigenesis. For example, tumor-derived exosomes (TEX) were shown to induce bone marrow progenitor cells differentiation into provasculogenic cells facilitating development of the pre-metastatic niche[54]. TEX can modulate immune response by delivering activated epidermal growth factor receptor (EGFR) to host macrophages rendering these cells less efficient in type I interferon production and compromising host antiviral immunity[55]. We previously showed that EVs contribute indirectly to immune suppression through expansion of T regulatory cells[56]. Moreover, recent studies reported that TEX carry PD-L1 molecules that contribute to immunosuppression and that PD-L1 levels on exosomes correlate with cancer activity and progression[57]. It was previously shown that OvCa exosomes can be transported from the periphery to the LNs in a process actively enhanced by lymphatic endothelial cells[58]. Also, a selective uptake of exosomes by various types of target T-cells has been described[54,59,60].

To the best of our knowledge we are the first to report ARG1 presence in tumor-derived EVs, although modulating activity of ARG1 in EVs has been recently reported in non-tumor settings. In a mouse model of the autoimmune disease alopecia areata, MDSC-derived exosomes containing ARG1 were preferentially taken up by activated T-cells and suppressed the proliferation, as well as CD3ζ and CD69 expression in activated LN cells and skin-infiltrating lymphocytes[61]. Similarly, exosomes with enzymatically active ARG1 that were isolated from granulocytic MDSCs of tumor-bearing mice attenuated DSS-induced murine colitis by suppressing Th1 cells and promoting Treg differentiation[62]. In diabetic mice, ARG1 was found to be enriched in serum exosomes and ARG1+ exosomes were effectively taken up by endothelial cell contributing to development of diabetic endothelial dysfunction resulting from inhibition of NO production[63]. Increased secretion of active ARG1 in exosomes was also observed after drug-induced liver injury[64]. ARG+ EVs secreted by hepatocytes were shown to modulate the blood metabolome associated with oxidative stress and endothelial regulation, and inhibited the acetylcholine-induced relaxation of isolated pulmonary arteries[64].

In light of these findings ARG1 carried by small EVs emerges as a potent metabolic and immunomodulatory factor, since it remains stable, may easily cross tissue barriers and quickly reaches secondary lymphoid organs. Vesicular ARG1 can act beyond the TME, being distributed in the peritoneal cavity by the ascites fluid and systemically through the peripheral circulation, leading to a systemic T-cell suppression. Therefore, although MDSCs and TAMs, in comparison to ARG1+-EVs, are the predominant sources of ARG1 within TME, ARG1 in EVs may exert systemic biological effects.

By using a small molecule arginase inhibitor, we have shown that the negative effects of this enzyme can be mitigated. Several previous studies indicated that blocking arginase activity is an attractive therapeutic target to promote anticancer immune responses. A number of synthetic as well as natural arginase inhibitors are under intense preclinical and clinical evaluation. Treatment with the small-molecule inhibitor nor-NOHA abrogated the arresting effects of arginase on T-cell proliferation and led to lymphocyte-dependent reduction of tumor growth[16], and similar effects were achieved in a murine OvCa model[40]. Arginase

inhibition also nearly completely abrogated the immunosuppressive effects of ARG2-positive circulating AML blasts[31]. Treatment with the small-molecule arginase inhibitor CB-1158 was shown to reduce tumor growth in several mouse models of cancer by increasing the number of tumor-infiltrating CD8+ T-cells and NK cells, and production of TH1-associated inflammatory cytokines[39]. CB-1158 also significantly improved the efficacy of checkpoint blockade (anti-PD1 and anti-CTLA-4), adoptive T-cell or NK cell therapy or chemotherapy with gemcitabine and is currently investigated in phase I clinical trial as a single agent or in combination with immune checkpoint therapy in patients with advanced or metastatic solid tumors (NCT02903914). Additionally, arginase activity was shown to impair CAR-T-cell therapy of neuroblastoma[30], emphasizing the significant clinical implications of arginase as a potential target for T-cell immunotherapy.

Altogether, we provide the first evidence for the role of ARG1 in the formation of an immunosuppressive microenvironment in OvCa. Moreover, we show that the effects of ARG1 are not only confined to the tumor site, but are disseminated through the release of ARG1-containing small EVs. These EVs transfer functionally active ARG1 as metabolic checkpoint molecules over a long distance to antigen-presenting cells and mitigate antitumor immune response, leading to an enhanced tumor growth in vivo. We identify hereby a novel mechanism of tumor-induced systemic T-cell dysfunction based on the activity of tumor-derived ARG1+ EVs, that may also apply to other arginase-expressing tumor types and may have significant clinical implications for T-cell immunotherapy approaches.

## Methods

**Reagents.** Recombinant human ARG1 was obtained from Biolegend (San Diego, CA, USA), recombinant human and murine IL-2, murine IL-4, and GM-CSF were purchased from Peprotech, arginase inhibitor OAT-1746 was synthesized at OncoArendi Therapeutics, Warsaw, Poland. All other reagents, if not otherwise stated, were obtained from Sigma-Aldrich.

**Cell lines.** Human ovarian cancer cell lines used in this study are listed in Supplementary Table 5. The murine ovarian cancer cell line ID8, derived from spontaneous malignant transformation of C57BL/6 MOSE cells in vitro was kindly provided by Kathy Roby from University of Kansas. For in vivo tumor models, VegfA and Defb29 transduced ID8 cells (ID8-VegfA/Defb29) kindly obtained from Jose R. Conejo-Garcia and Kathy Roby (University of Kansas Medical Center, KS, USA) were used. All cell lines were cultured in RPMI 1640 or in Dulbecco's modified Eagle's (DMEM) media supplemented with 10% (v/v) FCS, 2 mM L-glutamine, 100 U ml$^{-1}$ penicillin and 100 µg ml$^{-1}$ streptomycin at 37 °C in an atmosphere of 5% $CO_2$ in air. Tumor cell lines were regularly tested for Mycoplasma and confirmed to be negative.

**OvCa patient samples.** Primary ovarian carcinoma lesions were collected from 84 previously untreated patients with epithelial OvCa who were admitted to the Gynecologic Oncology Clinic at the University of Medical Sciences in Poznan, Poland. Histological diagnoses including tumor grade were determined by WHO criteria and were confirmed by a second review of the original H&E tissue sections. Normal ovarian tissues obtained from patients who underwent radical hysterectomy due to non-ovarian disease (e.g. uterine fibroids) were used as controls. Plasma arginase activity was measured in 81 OvCa patients and 10 normal controls (Supplementary Table 3). All OvCa patients underwent cytoreductive surgery and received subsequently standard first-line platinum-based chemotherapy. Ascites and/or plasma were collected during a routine medical procedure from 49 women diagnosed with stage III serous ovarian carcinoma at the Gynecologic Oncology Clinic at the University of Medical Sciences in Poznan, Poland or at the Department of Obstetrics and Gynecology of the Praski Hospital in Warsaw, Poland. In brief, 5–20 ml of ovarian cyst fluid was collected after removal of the cyst from the abdomen by puncturing the cyst wall with an 18-gauge needle mounted on a 10-ml syringe. Controls (n = 9), with histologically benign gynecological conditions including fibromas, endometriosis, and mucinous and serous cystadenomas were selected based on age-matching to patients with ovarian cancer. Normal control peripheral blood mononuclear cells (PBMC) were obtained by Histopaque-1077 (Sigma Aldrich) or Lymphoprep (Stemcell Technologies) separation from buffy coats from healthy volunteers, commercially obtained from the Regional Blood Centre in Warsaw, Poland. The approval for these studies was obtained from the Institutional Bioethical Review Board of the Medical University of Warsaw

(approval no. KB/117/2014) and from the University of Medical Sciences in Poznan (approval no. 214/11). All relevant ethical regulations for work with human participants were complied with and patients' samples were obtained according to the Declaration of Helsinki. Each patient signed a written informed consent for all the procedures.

**Isolation of EVs from supernatants and biological fluids**. Small EVs from cell culture supernatants were isolated by sequential centrifugation. Briefly, cells were grown in three-layer T-600 flasks in DMEM medium with exosome-depleted 5% FBS (Gibco Thermo Fisher Scientific) until they reached confluency of 85–95%. Next, the media (~100 ml) was collected and pre-cleared as follows: centrifugation at $500 \times g$ for 10 min at 4 °C, followed by a centrifugation at $2500 \times g$ for 20 min at 4 °C to remove cell debris and filtration through a 0.22 μm bacterial filter to remove larger microvesicles. The pre-cleared media was next ultracentrifuged at $110,000 \times g$ for 120 min (Optima XPN-100 ultracentrifuge, Beckman Coulter) using a Beckman Ti70 fixed-angle rotor. The EV pellet was washed with PBS followed by a second step of ultracentrifugation at $110,000 \times g$ for 1 h at 4 °C. EVs from human ascites (25–50 ml) or cyst fluid (10 ml) were isolated in a similar way except for the filtration step, which was replaced by a centrifugation at $10,000 \times g$ for 30 min at 4 °C. EVs from patients' plasma were isolated by SEC. Briefly, an aliquot of 500 μl plasma pre-cleared as described above was applied to a 10 ml qEV column (Izon Science). The exclusion volume fractions containing EVs were collected according to the manufacturer's instructions and concentrated by ultrafiltration using 100,000 MW spin columns (Millipore). For some applications tumor-derived EVs were isolated directly from ascites using immunomagnetic beads. Magnetic beads coated with anti-EpCAM (epithelial cell adhesion molecule) antibodies (Miltenyi) were incubated with 1 ml of ascites for 2 h at 4 °C with constant agitation. Then the magnetic immune complexes were applied onto a MS column and isolated according to manufacturer's instructions. Bound EVs were eluted with 100 μl of the recommended buffer, pelleted by centrifugation at $9000 \times g$ for 30 min. EVs from patient's cyst fluid or ascites used for functional assays were pre-enriched by sequential centrifugation as described above and then purified by SEC. Pelleted EVs used for Western Blot analyses were resuspended in RIPA lysis buffer supplemented with a protease inhibitors cocktail. EVs used for in vitro assays, flow cytometry analysis or electron microscopy were resuspended in 100 μl of PBS. Protein concentration of isolated EVs was determined by the BCA assay according to the manufacturer's instruction (Pierce BCA Protein Assay Kit, ThermoFisher Scientific).

**Characterization of isolated EVs**. EVs were evaluated for morphology by TEM and for particle distribution and size using tunable resistive pulse sensor (TRPS) technology (qNano instrument, Izon Science) or dynamic light scattering (Nano-Sight NS300 instrument, Malvern). For TEM the EV pellet obtained from 25 ml of ascites was resuspended in 50 μl of 2% paraformaldehyde, fixed for 1 h, washed in PBS, pelleted again by ultracentrifugation and resuspended in 20–50 μl of PBS. A 10 μl drop of the EV suspension was deposited directly on a Formvar-carbon-coated EM grid (Agar Scientific Ltd.) and left to absorb for 20 min. Then the grid was washed in PBS, fixed in 1% glutaraldehyde for 5 min and washed in PBS. Next, the sample was contrasted by putting the grid to a 50 μl drop of 4% uranyl-acetate solution pH 7 for 10 min and washed 7× in Millipore-grade water. The EVs were next embedded in a mixture of 2% uranyl acetate (UA) and 1% methyl cellulose (ratio of 100–900 μl, respectively) by placing the grid on a 50 μl drop of methyl cellulose–UA for 10 min on ice. Next, excess fluid was adsorbed and the grid was air dried for 5 min. For immunogold labeling EVs were fixed, adsorbed on a Formvar-carbon-coated grid as described before, and permeabilized by incubating in 0.05% saponin in PBS for 30 min. After washing 3× in 0.05% saponin in PBS the sample was blocked in 2% BSA/0.05% saponin in PBS for 10 min. Then, the grid was incubated on a 5 μl droplet of the primary antibody suspension (1:100) in 0.05% saponin in PBS for overnight. After washing three times in 0.05% saponin in PBS, antibody binding was detected by 1 h incubation with anti-rabbit IgG-conjugated 6 nm gold particles (1:200, Jackson ImmunoResearch) in blocking buffer. After washing 3× with Millipore H$_2$O the sample was contrasted and embedded as described above. The probes were analyzed with the use of TEM (JEM 1200 EX, JEOL Comp.). To optimize photomicrographs for printing, brightness, and contrast were adjusted with the use of Corel Photo-Paint 12. For particle analysis using the qNano system a NP150 nanopore and 70 nm calibration beads were used. For measurement the following settings were used: stretch: 46.23 mm, pressure: 10.0, voltage: 0.70 V. For nanoparticle tracking analysis a NS300 instrument equipped with a sCMOS camera, 488 nm laser, and NTA 3.0 0068 software was used. Per sample, 1500–2500 frames were analyzed using the following settings: camera level 14, camera gain 400, slider shutter 1000, and 25 frames/s. For both particle distribution and size analyses EV preparations were diluted 100–1000-fold in PBS for optimal analysis. The presence of typical exosomal tetraspanins on EVs was determined by flow cytometry. 10 μg of concentrated EVs pre-enriched by sequential centrifugation from ascites were coupled to magnetic beads coated with anti-CD9, anti-CD81, or anti-CD63, respectively, according to the manufacturer's instruction (ThermoFisher Scientific). Bead-bound EVs were collected, washed two times in PBS/0.1% BSA, stained with the appropriate fluorochrome-conjugated Abs and analyzed by flow cytometry. A single-beads gate was set based on the FCS and SSC scatter and a minimum of 3000 beads were acquired. We have submitted all

relevant data according to EV isolation and characterization to the EV-TRACK knowledgebase (EV-TRACK ID: EV190025)[65]. Our EV-metric is up to 78%.

**Immunohistochemistry**. Paraffin-embedded 5 μm sections of tumor tissues were deparaffinized, re-hydrated, and microwaved for 15 min in 0.01 M citrate buffer at pH 8.0 to retrieve antigens. Nonspecific binding was blocked with Protein Block, serum-free (Agilent Dako) for 45 min prior to the staining procedure. Following a 45 min incubation with an optimal dilution of primary Abs (dilutions of used Abs are listed in Supplementary Table 6) at RT in a moisture chamber, slides were washed in 0.5% (w/v) BSA and then incubated with secondary anti-rabbit or anti-mouse-Ab (Agilent Dako) coupled with horseradish peroxidase under the same conditions. The antigen–Ab complexes were visualized with 3,3′-diaminobenzidine (DAB) solution (1 mM DAB, 50 mM Tris–HCl buffer, pH 7.6) for 10 min. Tissue sections were counterstained with Mayer's hematoxylin solution and mounted in a mounting medium (Agilent Dako). The slides were evaluated by two independent investigators (E.E., S.C.). The percentage of negative, weakly, moderately, and strongly stained cells were evaluated as described in the captions for the Supplementary Table 2.

**Western blotting**. Lysed EVs or cells were sonicated 2× for 15 s in an ultrasonic water bath (BD ProbeTec ET). Subsequently, samples were boiled in Laemmli loading buffer, separated using SDS–PAGE and transferred to nitrocellulose membranes (Amersham) by semi-dry blotting. After blocking with 5% (w/v) non-fat dry milk in Tris-buffered saline/1% Tween-20 membranes were probed with primary antibodies followed by incubation with appropriate HRP-conjugated secondary antibodies. Antibodies used for immunodetection are listed in Supplementary Table 6. Bands were revealed using Western Bright Quantum Kit (Advansta). Stella Imaging System (Raytest Isotopenmessgeraete) or the Chemidoc Touch System (Bio-Rad) were used for image acquisition. ImageLab software version 5.2.1 (Bio-Rad) was used for densitometric analysis of the blots. For background subtraction, a rolling disc between 10 and 25 was used. Relative ARG1 expression was calculated by division of the optical density of the band of interest by the optical density of the band corresponding to 10 ng of recombinant human ARG1 (set as 1). Relative Tsg101 expression was calculated by division of the optical density of the band of interest by the optical density of the corresponding band of 10 μg of ID8-ARG1-V5 cell lysate. Beta-actin served as loading control. The uncropped images of all blots are shown in the Source Data file.

**Arginase activity assay**. The activity of arginase in OvCa patients' plasma was determined by measuring the conversion of L-arginine to L-ornithine in a colorimetric assay. Briefly, 2.5 μl MnCl$_2$ was added to 25 μl of plasma (10 mM end concentration) and the enzyme was activated at 37 °C for 5 min. Then, 50 μl of 0.5 M L-arginine-hydrochloride in 1% sodium carbonate/0.05% sodium bicarbonate buffer pH 10.5 was added (25 mM final concentration) and the samples were incubated for 40 min at 37 °C. The hydrolysis of L-arginine was stopped and the amount of produced L-ornithine was determined by adding 190 μl of 91% acetic acid/1.1% phosphoric acid/0.75% ninhydrin, and heating the samples for 15 min at 95 °C. After cooling of the samples to RT, absorbance was measured at 515 nm (Asys UVM 340 Plate Reader) and compared with L-ornithine monochloride standard curve. For the measurements of arginase activity in cells and EV fractions, a colorimetric assay for urea detection was used. For activity measurements in cell line supernatants $1 \times 10^4$ cells were incubated in 80 μl OptiMEM medium in a 96-well microplate in the presence of 20 mM L-arginine at 37 °C for 24 h. For activity measurements in EV lysates, 20 μg of patient-derived EVs were lysed in 1% (v/v) Triton X-100 in PBS in an end-volume of 50 μl. Then 50 μl of the cell supernatant or of lysed EVs, respectively, were added to 150 μl of freshly prepared mixture (1:1) of 4 mM o-phtaldialdehyde in 50 mM boric acid/1 M sulfuric acid/0.03% Brij-35, and 4 mM N-(1-Naphtyl) ethylene–diamine dihydrochloride in 50 mM boric acid/1 M sulfuric acid+/0.03% Brij-35 and incubated for 30 min at RT. Absorbance was measured at 540 nm (Asys UVM 340 Plate Reader).

**Generation of ID8 cells stably expressing murine ARG1**. ID8 cells stably expressing murine ARG1 tagged at C-terminus with V5-tag (ID8-ARG1-V5) were generated using lentiviral transduction. Murine full length *ARG1* cDNA was amplified from the pCMV6-Kan/Neo plasmid encoding murine ARG1 (OriGene Technologies) using the following primers: forward: 5′-GGCCGAATTCACC ATGAGCTCCAAAGCCAAAGTCC-3′ and reverse: 5′-GGCGGCGGCCGCTCA CGTA GAATCGAGACCGAGGAGAGGGTTAGGGATAGGCTTACCCTTAG GTGGTTTAAGGTAGTCAGTCC-3′ resulting in addition of C-terminal V5-tag. The tagged *ARG1* cDNA was inserted into the multicloning site of pLVX-IRES-Puro—the mammalian expression vector for bi-cistronic expression of a gene-of-interest, together with a puromycin-resistance marker (Thermo Fisher Scientific). Transduced ID8 cells were selected with 1 μg ml$^{-1}$ puromycin. ID8 cells transduced with the empty pLVX-IRES-Puro vector served as control cell line.

**EV endocytosis**. BMDCs precursors were isolated from bone marrow by flushing femur, tibia, and humerus bones of 6-week-old female C57BL/6 mice with cold PBS. After lysis of RBCs with ammonium chloride. Cells ($1 \times 10^7$ cells in 5 ml per well) were cultured in six-well plates in non-essential amino acid supplemented

RPMI 1640 medium with 10 ng ml$^{-1}$ granulocyte macrophage colony-stimulating factor (GM-CSF; PeproTech). On the second day, the culture medium was removed and fresh medium containing 10 ng ml$^{-1}$ GM-CSF and 5 ng ml$^{-1}$ interleukin (IL)−4 (PeproTech) was added. The procedure was followed by replacing 75% of medium with a fresh one containing 10 ng ml$^{-1}$ GM-CSF and 5 ng ml$^{-1}$ IL-4 on day 4. Loosely adherent cells were replated on day 6 into new six-well plates at a concentration of $5 \times 10^6$ cells in 5 ml per well in culture medium containing 10 ng ml$^{-1}$ GM-CSF and 5 ng ml$^{-1}$ IL-4. On day 7 of the culture cells were collected by gently scraping and incubated at a density of $5 \times 10^5$ cells in 500 µl medium with different amounts (1–100 µg) of EV-ARG1 for 1 h at 37 °C or on ice (negative control). Then the BMDC were washed with cold PBS to remove not endocytosed EVs and cytospun onto Superfrost plus slides at $300 \times g$ for 10 min at RT using Cytospin™ 4 Cytocentrifuge (Thermo Fisher Scientific). Subsequently the cells were fixed in 4% paraformaldehyde and mounted in DAPI-containing medium (Molecular Probes). The samples were visualized under a confocal microscope (Zeiss Axio Imager Z2 LSM 700) with the following parameters: ×63 magnification in oil, 488 nm laser, $P = 1.8$, gain = 657. Aliquots of the remaining samples were lysed in RIPA buffer for subsequent Western blot analysis.

**In vitro T-cell proliferation assay.** Freshly isolated PBMC were counted in Türk's solution and immediately used for experiments. CD8$^+$ and CD4$^+$ T cell subsets were separated using EasySep™ Human CD4$^+$ T-Cell Enrichment Kit or EasySep™ Human CD8$^+$ T-Cell Enrichment Kit (StemCell Technologies) according to the manufacturer's protocols. Purity of each fraction was >92%, with >98% viability. For cell proliferation assay, CD8$^+$ and CD4$^+$ T-cells were labeled with Cell Trace Violet dye (5 µM final concentration, Thermo Fisher Scientific) according to manufacturer's manual. Next, the labeled T-cells were plated in round-bottomed 96-well plates ($1–2 \times 10^5$ cell per well) in L-arginine-free RPMI-medium (SILAC RPMI-medium, Thermofisher) supplemented with 10% FBS, penicillin/strepto-mycin, 2% glutamine and 2% L-lysine, 150 µM L-arginine and stimulated with Dynabeads® Human T-Activator CD3/CD28 (Thermo Fisher Scientific) in the presence of 30 U ml$^{-1}$ recombinant human IL-2 (Peprotech). Patient-derived EVs pre-enriched by sequential centrifugation and purified by SEC were added in amounts corresponding to 2 ml of starting material (ascites or cyst fluid), if not other stated. The arginase inhibitor OAT-1746 was added as indicated in the figures. For some experiments T-cells were cultured in full ascites or in the ascitic supernatant after EVs isolation by sequential centrifugation (90% v/v). After incubation for 3–6 days at 37 °C, 5% CO$_2$ T-cells were harvested, stained with corresponding anti-CD3, anti-CD4, or anti-CD8 antibodies (Supplementary Table 6) and analyzed by flow cytometry (FACSCanto II, BD Biosciences). Per-centages of proliferating cells were calculated using the FlowJo Software v7.6.5 (Tree Star) and the percentage of proliferation inhibition in the presence of EVs was calculated relative to bead-stimulated control T-cells (Supplementary Fig. 8). Inhibitory potential of EV-ARG1 was evaluated in murine CD4$^+$ or CD8$^+$ T-cells isolated from spleens of 6-week-old C57BL/6 mice using EasySep™ Mouse CD8$^+$ T-Cell Isolation Kit or EasySep™ Mouse CD4$^+$ T-Cell Isolation Kit (StemCell Technologies). The proliferation assay set up was analogous to the one described above for the human T-cells, with exception for the use of Dynabeads Mouse T-Activator CD3/CD28 (Thermo Fisher Scientific) and anti-mouse CD3, CD4, or CD8 antibodies (Supplementary Table 6). For dendritic cells (DCs)-primed T-cell proliferation experiments, $5 \times 10^5$ murine BMDCs were incubated with 0.85–1 µg of EVs for 4 h at 37 °C in 500 µl full cell culture medium and then washed with cold PBS to remove not-internalized EVs. Additionally, for antigen-specific priming of CD8$^+$ OT-I cells $5 \times 10^5$ BMDCs were subsequently incubated with 1 µM SIIN-FEKL peptide (Sigma Aldrich) for 1 h after EVs wash out. Finally, $1 \times 10^5$ of BMDCs were added to murine CTV-stained CD8$^+$ or CD4$^+$ T-cells or to CD8$^+$ OT-I cells [1:1 ratio for Dynabeads®-induced proliferation and 2:1 ratio for antigen (SIINFEKL)-specific proliferation] and incubated for 72 h at 37 °C in round-bottomed 96-well plates in L-arginine-free RPMI medium. Where stated, ~100 µg of EVs were directly added to the T-cells.

**Flow cytometry analysis.** Flow cytometry was performed on FACSCanto II or Accuri flow cytometers (BD Biosciences) operated by FACSDiva 6.1.3 or BD Accuri C6 1.0 software, respectively. Gating strategies for flow cytometry are shown in Supplementary Figs. 9–12. For data analysis Flow Jo v7.6.5 software (Tree Star) was used. Fluorochrome-conjugated antibodies used for the stainings are listed in Supplementary Table 6. For cell surface staining, cells were blocked at RT in 5% normal rat serum in FACS buffer (PBS; 0.064% NaF) or 5% BSA in FACS buffer and then incubated for 30 min at RT with fluorochrome-labeled antibodies at dilutions determined by pre-titration including appropriate isotype controls. After washing in FACS buffer, cells were immediately analyzed. For intracellular staining cells were first stained with antibodies against cell surface markers and then fixed, permeabilized, and stained with the Intracellular Staining Kit (Affymetrix eBios-ciences) according to the manufacturer's instructions. Gating strategies for the respective experiments are shown in Supplementary Figs. 9–12.

**Mice.** Female 8-week-old to 14-week-old C57BL/6 mice were obtained from the Animal House of the Polish Academy of Sciences, Medical Research Center (Warsaw, Poland). C57BL/6-Tg(TcraTcrb)1100Mjb/J (OT-I) mice were purchased from the Jackson Laboratories. The experiments were performed in accordance with the guidelines approved by the first Local Ethics Committee of the University of Warsaw (approval no. 72/2014) and in accordance with the requirements of EU (Directive 2010/63/EU) and Polish (Dz. U. poz. 266/15.01.2015) legislation.

**In vivo T-cell proliferation assay.** OVA (SIINFEKL)-specific CD8$^+$ T cells were isolated from the spleen and LNs of OT-I mice, labeled with CTV (as described above) and transferred into the host C57BL/6 mice at a cell number of $2–3 \times 10^6$ in 150 µl of PBS. Twenty-four hours post OT-I T-cells inoculation, host mice were challenged with 5 µg of full-length OVA protein (grade VII, Sigma Aldrich) injected subcutaneously (s.c.) mixed with 100 µg of EV-ARG1 or control EV-pLVX in a total volume of 40 µl into the right thigh. Subcutaneous injections of 100 µg of EV-ARG1 or control EVs were repeated daily for the next 2 days. In some experimental settings 20 mg kg$^{-1}$ of the ARG inhibitor OAT-1746 was administered intraperitoneally (i.p.) twice daily, starting from the day of antigen challenge, for 3 following days. On day 3 post OVA immunization, T-cells from OVA injection site draining and contralateral inguinal LNs were harvested, stained with OVA-specific MHC tetramers (iTAg Tetramer/PE-H-2 Kb OVA (SIINFEKL), MBL Inc., WA, USA) to detect OT-I CD8$^+$ T-cells and analyzed for proliferation by flow cytometry.

**Ovarian tumor model.** Six-week-old to 8-week-old female C57/BL6 mice were injected i.p. with $4 \times 10^6$ ID8-VegfA/Defb29 cells lentivirally transduced with V5-tagged murine ARG1 (ID8-ARG1) or the control vector (ID8-pLVX) in 150 µl PBS. Tumor development was monitored by weight and waist measurements as signs of ascites formation. The treatment was initiated 2 weeks after tumor inoculation in the following scheme: 25 mg kg$^{-1}$ OAT-1746 twice daily i.p. for 1 week plus 12.5 mg kg$^{-1}$ OAT-1746 twice daily i.p. for the next 2 weeks. Control mice received i.p. PBS injections in the same regimen. Animals were euthanized when they gained more than 30% of weight and/or more than 40% waist circumference or showed any signs of cachexia or distress. For experiments aimed at immune cell pheno-typing mice were injected with $4 \times 10^6$ ID8-ARG1 or ID8-pLVX cells in 150 µl PBS and treated with 20 mg kg$^{-1}$ OAT-1746 twice daily i.p. for 2 weeks beginning on day 14 post tumor cells inoculation. Whole blood samples were obtained by facial vein bleed and allowed to clot for 1 h at RT. Serum ARG1 levels were measured in the sera by ELISA (LifeSpan BioSciences). Mice were sacrificed 2 weeks post treatment initiation, ascites fluid was collected, and the peritoneal cavity was washed with additional 5 ml of PBS. Cells and EVs from the collected ascites were used for Western blot analysis and immunophenotyping.

**Bioinformatics analysis.** Survival analysis according to ARG1 gene expression has been carried out for two transcriptomic data sets: (1) data set of 489 type II–IV serous ovarian cancers published by TCGA[66] and (2) Tumor Ovarian-Pamuła-Piłat-101-MAS5.0-u133p2 [GEO accession no. GSE63885] data set of 75 ovarian cancer tumors available through the R2 Genomics Analysis and Visualization Platform [an Affymetrix analysis and visualization platform developed in the Department of Oncogenomics at the Academic Medical Center at the University of Amsterdam (http://r2.amc.nl)]. OS analysis in the TCGA data set (1) has been carried out in R for the group of ovarian cancer patients based on primary tumor samples gene expression data. To make the analyzed patient group comparable to our IHC patient cohort according to age and ethnicity, the patient group has been filtered to exclude patients below 50 years of age and of other than white ethnicity. Subsets of the patients demonstrating either upper quartile Q3 ($n = 53$) or lower quartile Q1 ($n = 53$) of ARG1 expression level were compared. Significance of the observed effects has been determined using the Cox proportional hazards model with age, clinical stage, and tumor grade included in the analysis. In order to visualize the survival difference in R2 dataset (2), we used the log-rank test to find the point (cut-off) with the most significant (lowest P-value) split in high vs. low ARG1 level groups. Survival curves were derived by Kaplan–Meier method for these cut-offs.

**Statistical analysis.** Data are shown as means ± SD of at least three independent experiments. All analyses were done using GraphPad Prism 6.0 software (Graph-Pad Software Inc., La Jolla, CA, USA). Normal distribution of data was tested using the Shapiro–Wilk test. Differences between two groups were calculated using the unpaired two-tailed Student's t-test with Welch's correction or the nonparametric Mann–Whitney test for not normally distributed data. Statistical analyses of three or more groups were compared using one-way analysis of variance (ANOVA) followed by Bonferroni's multiple comparisons test. When the sample size was too small to test for normality, the Kruskal–Wallis test was used. $P < 0.05$ was considered statistically significant. Survival estimates were computed using the Kaplan–Meier plot and comparisons between groups were analyzed using the log-rank test.

## Data availability
Data are available within the article and supplementary files. The source data underlying Figs. 1a, e, 2a, b, d–g, 3a–e, 4b, 5d, 6a–d, 7a, b, Supplementary Figs. 2d, 3a–d, 5a, b, 6a, b, 7b, c, 8a are provided as a Source Data file. The TCGA microarray data are available in

the dbGaP database under the accession number PHS000178 and at the TCGA Data Portal (http://tcga.cancer.gov). The sequencing data of the Ovarian-Pamula-Pilat dataset are available in the GEO database under the accession number GSE63885. All other data that support the findings of the study are available from the corresponding author upon reasonable request.

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

## Acknowledgements

The work was supported by the grants 2013/11/B/NZ6/02790 and 2016/23/B/NZ6/03463 from the National Science Center, 692180 (STREAMH2020-TWINN-2015) from the European Commission Horizon 2020 Program, 265503/3/NCBIR/15 (STRATEGMED2) from the National Center for Research and Development, grant iONCO (Regionalna Inicjatywa Doskonałości) from the Ministry of Science and Higher Education in Poland, and NIH grants R01 CA 168628 and R21 CA205644.

## Author contributions

M.C.-K. designed and supervised the study, conducted the experiments, analyzed the data, and wrote the manuscript. A. Sosnowska performed in vivo studies, in vitro proliferation assays, and Western blot analysis. K.R. performed in vitro proliferation assays and Western blot analysis. D.N. designed the vector constructs, performed confocal microscopy, and wrote the manuscript. J.C.-T. participated in in vivo studies. M.S. supervised the collection of patient material and clinical data, performed tissue staining, and participated in EVs isolation. E.W. performed electron microscopy. P.G. performed bioinformatic analysis of expression data. M.G. participated in in vitro and in vivo experiments, and EV and PBMC isolation. Z.P. participated in in vivo experiments. A.Z. participated in in vitro experiments. K.S. performed Western blot analysis, arginase activity assays, and PBMC and EV isolations. A.G.-J. performed confocal microscopy. S.C., R.K., and E.E. prepared immunohistochemistry images and analyzed the tissue staining. S.G. performed statistical analyses. A. Stefanowicz collected blood and ascites samples. R.B., B.B., and A.G. designed and synthesized ARG1 inhibitor. T.W. participated in manuscript preparation and data interpretation and provided scientific advice. J.G. conceived the hypothesis, designed the study and wrote the manuscript. All authors edited and approved the final manuscript.

## Additional information

**Competing interests:** J.G. is a shareholder and Scientific Advisory Board member in OncoArendi Therapeutics, R.B., B.B., and A.G. are employees of OncoArendi Therapeutics, Warsaw, Poland. The remaining authors declare no competing interests.

