## [Peer Review File · Nature Communications]

Reviewers' comments:

Reviewer #1: Exosomes and cancer immunity
(Remarks to the Author):

This study shows for the first time that ovarian carcinoma tumor cell derived exosomes (TEX) contain ARG1, which leads to degradation of L-arginine in the tumor microenvironment. ARG1 has been shown to downregulate T cell Receptor complex and this impairs T cell function, and the authors show that OvCa derived TEX also have this effect. They show the clinical relevance in more than 80 patients, and also show the relevance in vivo in a mouse model where overexpression in the tumor leads to increased tumor growth in vivo. Also, injected EVs inhibit T cell activation in vivo.

It is a very well performed study, eventhough there are some figures which are shown as "one representative experiment out of 2" which is on the lower side.

ARG1 has already been shown to be present in exosomes, eventhough not from OvCa derived ones. MDSC-derived exosomes containing ARG1 has been reported to suppress T cell activation (ref 63 in the manuscript), so the novelty is OK but not very high.

Specific comments:

1. P.5, lines 133-137, "...and the rest..." Better to write the percentage out, a bit hard to interpret. Is it 56-33(one third)=11%?
2. Fig1a, has only one patient been tested, or is only one patient shown?
3. Fig 2b and described on p. 7, line182, the absence of CD63 in some samples should be discussed.
4. Fig 2d, far right, and text on line 200-201: the authors write "...higher ARG1 levels", but this is not significant.
5. It is not mentioned how the cyst fluid is sampled, but maybe this is a standard procedure?
6. Fig 3: dark blue and black are hard to distinguish from each other.
7. Fig. 4: I can't interpret the different amounts of TEX: what does 1 microliter of TEX correspond to in particles or protein? Is it 1 microliter of concentrated exosomes or original fluid?
8. Fig 5d is showing 1 out of 2 experiments, so I guess this is the mean+/-SD of what? Triplicates? This could use another experiment and rather to show SD between experiments.
9. Fig 6c is also showing "representative out of 2 experiments, perhaps better to write " one out of two with similar results, or make a third experiment.

Reviewer #2 :Exosomes
(Remarks to the Author):

The article by Czystowska-Kuzmicz et al describes an interesting novel finding, that human Ovarian Cancers express arginase1, which they secrete in extracellular vesicles, the latter inducing inhibition of T cell responses directly, or via capture by dendritic cells.

ARG1 has so far been described in immune cells, such as myeloid derived suppressor cells, rather than Tumor cells, thus this observation is very novel, and shows a new way for tumors to skew the immune responses.

Experiments are well performed, and interpretations are correct, and careful to avoid overinterpretations, which is extremely valuable. The authors combine observations in human ovarian cancer patients and samples, and a mouse model of ovarian cancer to provide some mechanistic studies.

I only have minor comments that could be considered to improve the article :

- 1) the authors did follow rather well the guidelines on EV studies (Lötvald et al, J Extracell Vesicles 2014), however, their interpretation that the ARG1 containing vesicles are specifically exosomes (of endosomal origin) is not really demonstrated here. The authors show random correlation of expression of ARG1 and TSG101 or CD63 by WB in ascites extracellular vesicles (fig2b), and

complete immunoisolation of ARG1 with EpCAM-positive vesicles whereas additional CD63+/EpCAM- vesicles do not contain ARG1 (suppl fig2C). Without demonstration that EpCAM is in majority absent from the plasma membrane and mainly inside internal compartments, it is possible that the EpCAM+ EVs in fact originate from the PM. Since the intracellular origin of the ARG1+ vesicles is not really important for the message, I would suggest to refrain from using the term exosomes, and rather prefer a nomenclature like EpCAM+ small EVs (EpCAM-sEVs) throughout the paper. If they choose to call these EVs exosomes, the authors must clearly specify upfront what they define under this term, ie not specifically endosome-derived sEVs.

2) To strengthen the message of ARG1 release in EVs, and although the authors show that soluble recombinant ARG1 does not display the immunosuppressive activity of EVs, it is important to determine whether ARG1 is also released as a soluble form from cells, in which case a) what proportion is in EVs versus soluble, and b) is the soluble supernatant devoid of EVs displaying, or not, immunosuppressive activity.

3) although the in vivo mouse model is necessary to provide some in vivo mechanistic information, it relies on overexpression of a tagged ARG1-V5 protein. This overexpression may lead to artificial release of ARG1 outside the cells, which would not occur in cells that do not overexpress the molecule : can the authors determine whether the level of expression of ARG1 in the ID8-transfected cells is similar to levels observed spontaneously in human OvCa or much higher ?

4) for discussion : How would ARG1 contained inside EVs (presumably) be released outside or inside target cells (T cells, DCs) to be active ? Is fusion or destruction of the vesicles required ? Also, it is not clear whether ARG1 needs to be transferred to target cells to be immunosuppressive or if it acts extracellularly. Finally, Fig5c shows even stronger inhibition of T cell activation by ARG1 inhibitor in the absence than presence of DCs. So what is the actual model for the mode of action of released ARG1: does it target the extracellular milieu, or T cells, or DCs ?

5) Technical points :

Western Blots should show the position of Molecular Weight markers for all proteins analysed, and full images as supplementary data.

Figure 4 shows suppressive activity of EVs from 2 patients: the authors should indicate from how many patients in total they performed this experiment, and from how many they observed this suppressive effects (or show quantification of inhibition in more than one patient).

The authors should consider submitting their experimental set up to the EV-TRACK data base (www.evtrack.org) and provide the code number. The calculated EV-TRACK index will most likely be relatively high.

RESPONSEs TO REFEREEs

Reviewer #1:

Specific comments:

1. P.5, lines 133-137, "...and the rest..." Better to write the percentage out, a bit hard to interpret. Is it 56-33(one third)=11%?

Answer: We now provide more detailed information on the number of patients and percentages (see page 5, lines 133-136 in the revised manuscript).

2. Fig1a, has only one patient been tested, or is only one patient shown?

Answer: One patient is shown – using immunoblotting we have measured arginase-1 in a single patient only. We have first observed that arginase-1 is expressed in established ovarian cancer cells, and wanted to confirm this finding in a primary material. However, after analyzing a single patient sample, we preferred to focus on immunohistochemistry as this is a standardized procedure allowing also to see which cells produce arginase-1. Thus, as shown in the manuscript, we have done immunohistochemistry in a total of 84 patients.

3. Fig 2b and described on p. 7, line182, the absence of CD63 in some samples should be discussed.

Answer: Indeed, the CD63 tetraspanin was not detectable in Western blotting in all exosome samples isolated from patients' ascites. There are several reports showing that CD63 is often absent or less abundant than other tetraspanins in exosome preparations from tumor cell lines or from biological fluids of tumor patients, including ascites [J Extracell Vesicles. 2013;2:20424; PNAS 2016;113(8):E968-77; Mol Cancer Res. 2017; 15(1):78-92]. Furthermore, for some cancer types a decrease in CD63 expression on tumor cells has been linked to their invasive and metastatic ability and to tumor progression [Onciol. Lett. 2017;13(6):4245-4251. Lung Cancer 2007;57:46–53; Lab Invest. 2002;82:1715–1724; Int J Cancer 2015;136:2304–2315;]. Since our analyzed exosome preparations from ascites were isolated from cancer patients at various stages of disease including advanced ovarian cancer, one can expect a similar downregulation of CD63 in some patient samples. Furthermore, not all antibodies dedicated for immunoblotting applications and working well with cell lysates are suitable for the detection of exosomal proteins. Since we had only limited amounts of exosome material from patients, we were not able to assess the suitability of other antibodies for Western blot detection. However, prompted by the reviewer's concern, we performed additional detailed comparative analysis of 3 typical surface-markers in our preparation of

exosomes – the tetraspanins CD9, CD81 and CD63 by immune-isolation and flow cytometry – a technique that requires less material than the Western blotting approach. We used three different bead types –coated with antibodies targeting either CD9, CD63 or CD81 and analyzed exosomes pre-enriched by size-exclusion chromatography from ascites of 6 different OvCa patients. Our analysis shows, that although many of the ascites-derived exosome samples bear all three tetraspanins, there are also exosome samples, which are positive only for selected tetraspanins. We have included these results in the supplementary material section and added appropriate text in the manuscript (page 7, lines 184-190 in the revised manuscript).

4. Fig 2d, far right, and text on line 200-201: the authors write “..higher ARG1 levels”, but this is not significant.

Answer: We have corrected this sentence to clearly indicate: “[...] slightly higher, but statistically insignificant ($P=0.0512$) ARG1 levels” (see page 8; line 208).

5. It is not mentioned how the cyst fluid is sampled, but maybe this is a standard procedure?

Answer: Indeed, this a standard medical procedure – 5 to 20 mL of ovarian cyst fluid was collected after removal of the cyst from the abdomen by puncturing the cyst wall with an 18-gauge needle mounted on a 10-ml syringe. We now provide this information in the methods section (page 20; line 512, and page 21; lines 515-517).

6. Fig 3: dark blue and black are hard to distinguish from each other.

Answer: We have changed the colors. Indeed, the data are better distinguished now.

7. Fig. 4: I can't interpret the different amounts of TEX: what does 1 microliter of TEX correspond to in particles or protein? Is it 1 microliter of concentrated exosomes or original fluid?

Answer: Since the protein concentration of isolated exosomes differed between patients and even more between patients and NC, we decided to relate the amount of added exosomes used in functional assays to the volume of the starting material, namely ascites or cyst fluid. This allowed us to make direct comparisons of patients and NC. In most assays (if not stated otherwise) we used exosome amounts corresponding to 2 ml of starting material. In Fig. 4c the 8 μ l of TEX corresponds to 2 ml of ascites, 4 μ l corresponds to 1 ml, 2 μ l corresponds to 0.5 ml of ascites and 1 μ l corresponds to 0.25 ml of ascites, respectively. In Fig. 4d the amount of added exosomes corresponds to 2 ml ascites. We now provide more detailed information on this issue in the description to Fig. 4 (page 48, lines 1216-1219 in the revised manuscript).

8. Fig 5d is showing 1 out of 2 experiments, so I guess this is the mean+/-SD of what? Triplicates? This could use another experiment and rather to show SD between experiments.

9. Fig 6c is also showing “representative out of 2 experiments, perhaps better to write “one out of two with similar results, or make a third experiment.

Answer to 8 and 9: We have corrected descriptions to these two figures as well to other where a similar point might be raised and provide the information that one out of two experiments with similar results is shown (page 50, lines 1242-1243, and page 52; line 1274 in the revised manuscript).

Reviewer #2: I only have minor comments that could be considered to improve the article:

1) the authors did follow rather well the guidelines on EV studies (Lötvall et al, J Extracell Vesicles 2014), however, their interpretation that the ARG1 containing vesicles are specifically exosomes (of endosomal origin) is not really demonstrated here. The authors show random correlation of expression of ARG1 and TSG101 or CD63 by WB in ascites extracellular vesicles (fig2b), and complete immunoisolation of ARG1 with EpCAM-positive vesicles whereas additional CD63+/EpCAM- vesicles do not contain ARG1 (suppl fig2C). Without demonstration that EpCAM is in majority absent from the plasma membrane and mainly inside internal compartments, it is possible that the EpCAM+ EVs in fact originate from the PM. Since the intracellular origin of the ARG1+ vesicles is not really important for the message, I would suggest to refrain from using the term exosomes, and rather prefer a nomenclature like EpCAM+ small EVs (EpCAM-sEVs) throughout the paper. If they choose to call these EVs exosomes, the authors must clearly specify upfront what they define under this term, ie not specifically endosome-derived sEVs.

Answer: This comment is related to the comment of the other Reviewer – please see our answer to point 3 above. Altogether, considering that we have used size exclusion chromatography, which is a preferred method to isolate exosomes, we show correct size of the vesicles, we use electron microscopy, Western blotting (albeit with somehow poor staining for CD63) and flow cytometry analysis for tetraspanins routinely used to identify exosomes, we decided to use the term exosomes, wherever our analyses confirm this.

2) To strengthen the message of ARG1 release in EVs, and although the authors show that soluble recombinant ARG1 does not display the immunosuppressive activity of EVs, it is important to determine whether ARG1 is also released as a soluble form from cells, in which case a) what proportion is in EVs versus soluble, and b) is the soluble supernatant devoid of EVs displaying, or not, immunosuppressive activity.

Answer: ARG1 is certainly also released as a soluble form from cells. This has been shown for myeloid cells, which infiltrate the tumors. We are not aware of any method that would allow us to calculate in a precise quantitative manner the amount of arginase in soluble and exosomal fractions. First, we have no guarantee that even if we isolate exosomes from ascites the remaining ascitic fraction is completely devoid of exosomes. Moreover, the isolation procedure is skewed by a certain loss of exosomes. Nonetheless, we have done additional experiments to address the points of Reviewer. First we have measured arginase activity in the full ascites and in the ascitic fraction after isolation of exosomes. These experiments have clearly shown that removal of exosomes reduces arginase activity in all investigated samples (see a new figure in the manuscript - Supplemental Fig. 5a and lines 241-244 on pages 9 and 10 of the revised manuscript).

Then, we have incubated human CD8⁺ T-cells with ascites. In parallel, we have isolated exosomal fraction from the same ascites and used both exosomes as well as exosome-depleted ascites to see the effects of all these fractions on the proliferation of CD8⁺ T-cells triggered with anti-CD3/anti-CD28 beads (see Supplemental Fig. 5b).

Aware of all these issues we did the in vivo experiments with recombinant arginase making us more convinced that the soluble form of the enzyme is probably more important locally, at site of its release. However, considering that T-cell proliferation occurs rather in distant lymph nodes, we might speculate that the effective way of delivering the enzyme is by means of extracellular vesicles.

u

3) although the in vivo mouse model is necessary to provide some in vivo mechanistic information, it relies on overexpression of a tagged ARG1-V5 protein. This overexpression may lead to artificial release of ARG1 outside the cells, which would not occur in cells that do not overexpress the molecule : can the authors determine whether the level of expression of ARG1 in the ID8-transfected cells is similar to levels observed spontaneously in human OvCa or much higher?

Answer: We have done additional in vivo experiment to address this comment. To this end we have inoculated mice (n=7) with ID8-transfected cells and collected ascites as they formed at days 24 and 34 of the experiment. The mean ARG1 activity in these mice was 2.08 mU/mL, which is within the range of concentrations measured in ovarian cancer patients (0.305 mU to 6.514 mU per ml of ascites, see the manuscript – page 8; line 214). The information on arginase activity in murine model is now provided in the manuscript (page 13; lines 333-337).

4) for discussion: How would ARG1 contained inside EVs (presumably) be released outside or inside target cells (T cells, DCs) to be active ? Is fusion or destruction of the vesicles required ? Also, it is not clear whether ARG1 needs to be transferred to target cells to be immunosuppressive or if it acts extracellularly. Finally, Fig5c shows even stronger inhibition of T cell activation by ARG1 inhibitor in the absence than presence of DCs. So what is the actual model for the mode of action of released ARG1: does it target the extracellular milieu, or T cells, or DCs ?

Answer: These are all very important questions. We can only speculate that exosomes are taken up by DCs by endocytosis. What happens afterwards is even less clear, and while we have been considering various potential mechanisms of arginase incorporation, we are currently unable to study this. ARG inhibitor that we use easily penetrates cell membranes and inhibits both the intracellular as well as the released enzyme. We decided to refrain from addressing these issues in the manuscript too avoid excessive speculations.

5) Technical points:

Western Blots should show the position of Molecular Weight markers for all proteins analysed, and full images as supplementary data.

Answer: Together with all figures we submit full images of original Western blots. In some cases the blotting membranes were cut with scissors to detect proteins of low and high molecular mass after a single electrophoresis – in these cases we show all parts of the membranes (see Supplemental Fig. 13-20).

Figure 4 shows suppressive activity of EVs from 2 patients: the authors should indicate from how many patients in total they performed this experiment, and from how many they observed this suppressive effects (or show quantification of inhibition in more than one patient).

Answer: Data from 2 patients refer only to part (d) of Figure 4. The suppressive effects of EVs are shown in part (b) and show cumulative data from 40 patients. With 2 of them we have done additional experiments investigating the effects of arginase inhibitor (OAT-1746).

The authors should consider submitting their experimental set up to the EV-TRACK data base (www.evtrack.org) and provide the code number. The calculated EV-TRACK index will most likely be relatively high.

Answer: We have submitted our experimental set up to the EV-TRACK database. However, since we have done additional analyses of our exosomal fraction (see the results with anti-CD9, anti-CD81 and antiCD63 beads in flow cytometry) we have not yet obtained the code number and we are just a couple of days before a standard 3-month period for resubmission of our revised manuscript.

Reviewers' comments:

Reviewer #1 (Remarks to the Author):

The manuscript is now considerably improved, the authors replied to all my comments and I recommend publication.

Reviewer #2 (Remarks to the Author):

In this revised version, the authors have provided new experiments, some of which are interesting and strengthen the paper (figS5). However, I am still not satisfied with their choice of the term exosomes, and their justification for that. They misleadingly quote the 2014 guidelines of ISEV (Lotvall et al, JEV 2014) to claim that their vesicles fit the definition of exosomes, whereas the Lotvall paper does not provide any definition of exosomes, and instead specifies that no specific markers of exosomes exists and supports the use of the term EV instead. The size and spherical shape described for the EVs analysed here is shared by exosomes and small plasma membrane-derived EVs, thus cannot be used as a proof of exosome nature.

The new experiments where the ascites EVs are immuno-isolated by antibodies to a given tetraspanin (CD9, CD81 or CD63) and labeled with antibodies to the same or the other tetraspanins (new fig S2) are interesting but only show the heterogeneity of small EVs recovered from ascites, some patients ascites containing EVs bearing 2 or 3 and some others bearing some single tetraspanin+ EVs. Certainly, the authors do in most cases probably recover actual exosomes in their EV isolates (if one assumes that exosomes are CD63+/CD9+/CD81+ triple positive EVs, as proposed by Kowal et al, PNAS 2016, which is not even proven in other models). However none of the experiments of this article (including the new ones), is demonstrating that these particular multiTSPAN+ EVs/exosomes are the ones containing arginase, rather than other small EVs forming at the plasma membrane. Since the authors clearly show that ARG1 is in EpCAM+ EVs, which may or may not be endosome-derived exosomes, and not in EpCAM-/CD63+ EVs, which could also or instead be exosomes (figS2), I do not understand their insistence on using a term that is not appropriate in a situation where the actual origin of their EVs is not a major point of their study. The authors MUST call their vesicles in the paper sEVs, possibly EpCAM+ small EVs. And they must amend their text in p7-p8 to only describe the features of their EVs (size, aspect by EM) without claiming that they are typical of exosomes. A minima, they must delete the term exosomes from the title and the abstract, use extracellular vesicles instead, and in the main text of the paper, after amending the paragraphs p7-8, they can specify that, even though they do not conclusively demonstrate here that the ARG1+ EVs are exosomes, they choose to use this term for the rest of the paper, applying it to any type of small EVs.

On the other hand, the authors provide an interesting new experiment shown in figS5, whereby they convincingly show that the suppressive activity of ascites is clearly depleted by the ultracentrifugation process that eliminates a large part of EVs, and recovered in the corresponding EVs (although it would have been better to show the suppressive activity of EVs recovered from the same volume of ascites pre-post-depletion, in addition to the EVs from 2ml of ascites).

the answer about EV-TRACK is surprising: in my experience, EV-TRACK provides a reference number immediately upon submission of the technical data of a paper, thus the authors must be able to indicate this number in the M&M section.

RESPONSEs TO REFEREEs

Reviewer #2:

In this revised version, the authors have provided new experiments, some of which are interesting and strengthen the paper (figS5). However, I am still not satisfied with their choice of the term exosomes, and their justification for that. They misleadingly quote the 2014 guidelines of ISEV (Lotvall et al, JEV 2014) to claim that their vesicles fit the definition of exosomes, whereas the Lotvall paper does not provide any definition of exosomes, and instead specifies that no specific markers of exosomes exists and supports the use of the term EV instead. The size and spherical shape described for the EVs analysed here is shared by exosomes and small plasma membrane-derived EVs, thus cannot be used as a proof of exosome nature.

The new experiments where the ascites EVs are immuno-isolated by antibodies to a given tetraspanin (CD9, CD81 or CD63) and labeled with antibodies to the same or the other tetraspanins (new fig S2) are interesting but only show the heterogeneity of small EVs recovered from ascites, some patients ascites containing EVs bearing 2 or 3 and some others bearing some single tetraspanin+ EVs. Certainly, the authors do in most cases probably recover actual exosomes in their EV isolates (if one assumes that exosomes are CD63+/CD9+/CD81+ triple positive EVs, as proposed by Kowal et al, PNAS 2016, which is not even proven in other models). However none of the experiments of this article (including the new ones), is demonstrating that these particular multiTSPAN+ EVs/exosomes are the ones containing arginase, rather than other small EVs forming at the plasma membrane. Since the authors clearly show that ARG1 is in EpCAM+ EVs, which may or may not be endosome-derived exosomes, and not in EpCAM-/CD63+ EVs, which could also or instead be exosomes (figS2), I do not understand their insistence on using a term that is not appropriate in a situation where the actual origin of their EVs is not a major point of their study.

The authors MUST call their vesicles in the paper sEVs, possibly EpCAM+ small EVs. And they must amend their text in p7-p8 to only describe the features of their EVs (size, aspect by EM) without claiming that they are typical of exosomes. A minima, they must delete the term exosomes from the title and the abstract, use extracellular vesicles instead, and in the main text of the paper, after amending the paragraphs p7-8, they can specify that, even though they do not conclusively demonstrate here that the ARG1+

EVs are exosomes, they choose to use this term for the rest of the paper, applying it to any type of small EVs.

On the other hand, the authors provide an interesting new experiment shown in figS5, whereby they convincingly show that the suppressive activity of ascites is clearly depleted by the ultracentrifugation process that eliminates a large part of EVs, and recovered in the corresponding EVs (although it would have been better to show the suppressive activity of EVs recovered from the same volume of ascites pre-post-depletion, in addition to the EVs from 2ml of ascites).

The answer about EV-TRACK is surprising: in my experience, EV-TRACK provides a reference number immediately upon submission of the technical data of a paper, thus the authors must be able to indicate this number in the M&M section.

Answer: We would like to thank the Reviewers for very critical reading and constructive comments that improve our manuscript. The rationale for our stubborn use of the term exosomes stemmed from the fact that we tried to follow previous findings (refs 32 and 33 in the manuscript) reporting that extracellular vesicles isolated from ovarian carcinoma exert immunoregulatory effects. The authors of these reports have specifically referred to nanometer-sized membrane-encapsulated extracellular vesicles as exosomes. We tried to follow the protocols used in those articles and thus have used the same term. However, we thoroughly understand and agree with the arguments of our Reviewer. Thus, throughout the manuscript, we now use the terms: small extracellular vesicles, extracellular vesicles or tumor-derived extracellular vesicles (the latter used for EVs isolated from established tumor cell lines). We left the term exosomes in those instances, where we cite the articles that have specifically used this term.

We also provide the EV-TRACK number, which is EV190025. Our EV-metric is up to 78%.

We have also introduced few minor changes in the text (some typos, commas, etc). All of these are indicated in red color of the font.